# Hepatitis B virus promotes liver cancer by modulating the immune response to environmental carcinogens

Mei Huang [1,2,3,8], Dongyao Wang [1,2,4,8], Jiao Huang [1,2], An-Na Bae [5], Yun Xia [1,2], Xutu Zhao [1,2], Mahsa Mortaja[1,2], Marjan Azin [1,2], Michael R. Collier[6], Yevgeniy R. Semenov [6,7], Jong Ho Park [1,2,5] ✉ & Shadmehr Demehri [1,2] ✉

Hepatitis B virus (HBV) infection is associated with hepatitis and hepatocellular carcinoma (HCC). Considering that most HBV-infected individuals remain asymptomatic, the mechanism linking HBV to hepatitis and HCC remains uncertain. Herein, we demonstrate that HBV alone does not cause liver inflammation or cancer. Instead, HBV alters the chronic inflammation induced by chemical carcinogens to promote liver carcinogenesis. Long-term HBV genome expression in mouse liver increases liver inflammation and cancer propensity caused by a carcinogen, diethylnitrosamine (DEN). HBV plus DEN-activated interleukin-33 (IL-33)/regulatory T cell axis is required for liver carcinogenesis. Pitavastatin, an IL-33 inhibitor, suppresses HBV plus DEN-induced liver cancer. IL-33 is markedly elevated in HBV⁺ hepatitis patients, and pitavastatin use significantly correlates with reduced risk of hepatitis and its associated HCC in patients. Collectively, our findings reveal that environmental carcinogens are the link between HBV and HCC risk, creating a window of opportunity for cancer prevention in HBV carriers.

Hepatitis B virus (HBV) is a hepatotropic DNA virus that can cause persistent infection in primates[1–3]. It is estimated that the prevalence of HBV infection is around 250 million cases worldwide and is associated with an increased risk of hepatitis and HCC[4,5]. HBV's high prevalence has led to the development of a vaccine using hepatitis B surface antigen (HBsAg)[6,7]. However, HBV remains a significant global health threat. Key contributing factors include limited access to the HBV vaccine in certain countries, noncompliance with multi-dose regimens, and the presence of vaccine non-responders, estimated at 16% of the population, which are more prevalent among older individuals, those

with obesity, and smokers[8–11]. Importantly, these same risk factors have well-known carcinogenic effects and may be the primary drivers of hepatitis and HCC in HBV-infected liver[12,13]. As such, it is essential to determine the underlying mechanism linking HBV infection to hepatitis and HCC in order to effectively address this global challenge.

HBV DNA can integrate into the host genome, which has been proposed as a mechanism for genomic mutations with oncogenic effects[14]. However, HBV is typically a non-cytopathic virus programmed for benign episomal replication[3]. Although 10-30% of HCC cases are associated with HBV infection, the HBV genome integration

[1]Center for Cancer Immunology, Krantz Family Center for Cancer Research, Massachusetts General Hospital and Harvard Medical School, Boston, MA, USA. [2]Cutaneous Biology Research Center, Department of Dermatology, Massachusetts General Hospital and Harvard Medical School, Boston, MA, USA. [3]Department of General Surgery, The First Affiliated Hospital of USTC, Division of Life Sciences and Medicine, University of Science and Technology of China, Hefei, China. [4]Department of Hematology, The First Affiliated Hospital of USTC, Division of Life Sciences and Medicine, University of Science and Technology of China, Hefei, China. [5]Department of Anatomy, School of Medicine, Keimyung University, Daegu, South Korea. [6]Department of Dermatology, Massachusetts General Hospital and Harvard Medical School, Boston, MA, USA. [7]Laboratory of Systems Pharmacology, Harvard Program in Therapeutic Science, Harvard Medical School, Boston, USA. [8]These authors contributed equally: Mei Huang, Dongyao Wang. ✉e-mail: jpark@dsmc.or.kr; sdemehri1@mgh.harvard.edu

ratio in tumor cells is lower than in non-tumor tissues[15,16]. The other proposed mechanism linking HBV to HCC relates to HBV's ability to induce chronic inflammation in the liver[17,18]. HBV infection tightly interacts with the host immune system[3,19–21]. CD8[+] T cell response to HBV infection, as a potential therapeutic target, can lead to viral clearance but may paradoxically contribute to disease pathogenesis during the early stages[22,23]. CD8[+] T cells' attempt to eliminate infected hepatocytes and HBV-associated HCC cells can instead result in T cell exhaustion, resulting in chronic inflammation and the development of malignancy[24,25]. The impaired function of CD4[+] T cells and natural killer cells is another significant consequence of HBV infection[26,27]. In addition, the elevated neutrophil/lymphocyte ratio in peripheral blood and increased number of Foxp3[+] regulatory T cells (Tregs) are positively correlated with disease progression in patients with HBV-associated HCC[4]. However, there is limited inflammation in the 'immune tolerant' and 'immune inactive' stages of chronic HBV infection[28–30]. While immune imbalance induced by chronic HBV infection is considered the cause of HBV-related HCC development[31], the factors that induce this immune imbalance are unclear. In fact, HBV infection often does not cause any overt liver pathology, and up to 60% of HBV-infected individuals remain asymptomatic[32]. The combination of HBV infection and environmental factors, such as alcohol and smoking, is associated with increased liver injury[33–35]. This suggests that HBV alone may not be sufficient to induce chronic inflammation or liver cancer. Although the incidence of HBV infection is decreasing owing to vaccination, HBV infection remains a serious health problem globally[10,36]. Thus, it is essential to determine the full biological impact of HBV infection on the liver to uncover suitable therapeutic agents for hepatitis and HCC prevention in HBV-infected individuals.

Herein, we investigated the mechanism linking HBV to chronic hepatitis and HCC development. Importantly, mice chronically inoculated with the HBV genome did not develop liver inflammation or cancer. Instead, HBV modulated the liver response to a tobacco smoke carcinogen, diethylnitrosamine (DEN)[37,38], resulting in severe liver cancer phenotype. HBV-containing liver exposed to DEN upregulated interleukin-33 (IL-33), which was required for liver cancer development. Loss of IL-33 markedly reduced IL-33 receptor (IL-1 receptor-like 1 (IL1RL1) or ST2) expressing Tregs in HBV+DEN-treated liver. Accordingly, ST2 deletion on Tregs blocked liver cancer development in HBV+DEN-treated mice accompanied by increased cytotoxic CD8[+] T cells in the liver. Treatment with pitavastatin, an inhibitor of IL-33 expression[39], resulted in reduced risk of chronic hepatitis and liver cancer in mice and humans. Our findings indicate that HBV-infected individuals can be protected from hepatitis and HCC by minimizing their risk of exposure to chemical carcinogens. Furthermore, suppressing IL-33/Treg axis in the liver using statins represents a safe and effective strategy for the prevention and treatment of chronic hepatitis and its cancer sequelae.

## Results

### HBV augments carcinogen's effect to promote liver cancer

To establish an HBV infection-like model in mice, we employed a pAAV-HBV1.2 vector to deliver the HBV genome to hepatocytes via hydrodynamic tail vein injection[40,41]. This method resulted in persistent HBV gene expression in the liver, as evidenced by detectable HBsAg in the circulation for 6 months post-infection (Supplementary Fig. 1a). Importantly, HBV genome expression in 28-day-old wild-type (WT) mice did not cause any liver cancer over 12 months post-infection (Fig. 1a, b). In contrast, HBV-expressing WT mice that received a carcinogen, DEN, (HBV+DEN) showed a significantly higher liver tumor burden, reduced survival, and increased PCNA[+] proliferating hepatocytes compared with the Sham+DEN group (Fig. 1a–f and Supplementary Fig. 1b, c). Importantly, HBV DNA and gene expression persisted in HBV+DEN and HBV+phosphate-buffered saline (PBS)-treated WT livers up to 8 months post-infection (Supplementary

Fig. 1d–g). Thus, the combination of HBV and carcinogen accelerates liver cancer development, whereas HBV alone does not cause hepatocyte proliferation or cancer.

### DEN causes severe chronic inflammation in HBV-expressing liver

To explore the link between chronic hepatitis and liver cancer development in HBV-positive mice, we examined the immune environment of the liver at 4 months post-infection prior to any cancer development (Fig. 1a, b). HBV genome expression alone did not cause any liver damage measured by serum alanine aminotransferase (ALT) levels or liver inflammation assessed by CD45[+] leukocyte counts in the liver (Fig. 2a, b, and Supplementary Fig. 2a). However, CD45[+] leukocyte counts were significantly increased in the HBV+DEN group compared with the Sham+DEN, HBV+PBS, and Sham+PBS groups (Fig. 2a, b). Additionally, HBV+DEN-treated mice exhibited elevated serum ALT levels compared with HBV+PBS and Sham+PBS-treated mice (Supplementary Fig. 2a). To investigate the underlying mechanism leading to more severe chronic liver inflammation in HBV+DEN compared with Sham+DEN-treated mice, we performed whole liver RNA sequencing at 4 months post-infection. Gene Set Enrichment Analysis (GSEA) revealed that the HBV+DEN-treated liver upregulated hepatitis-related genes compared with Sham+DEN group (Fig. 2c). Further analysis identified six differentially expressed cytokines: *Il33*, *Il1b*, *Il4*, *Il16*, *Il7*, and *Il18* (Fig. 2d). Previous studies have linked IL-33 to cancer-prone chronic inflammation[39,42,43], prompting us to explore its role in HBV+DEN-associated liver cancer development. HBV+DEN-treated mice exhibited higher IL-33 expression in their liver compared with Sham+DEN-treated mice, which was expressed by E-cadherin[+] hepatocytes and α-SMA[+] hepatic stellate cells (Fig. 2e–g and Supplementary Fig. 2b). By contrast, IL-1β, IL-4, and IL-16 protein levels were not increased in HBV+DEN- compared with Sham+DEN-treated WT liver (Supplementary Fig. 2c-e). We have demonstrated that the Toll-like receptor (TLR) 3-4/TBK1/IRF3 pathway drives IL-33 expression in chronic inflammation[39]. Accordingly, activated TBK1 (phospho-TBK1) and IRF3 (phospho-IRF3) were induced in HBV+DEN-treated liver compared with the other treatment groups (Supplementary Fig. 3a). DEN induces immunogenic cell death in hepatocytes[44,45]. Interestingly, HBV+DEN treatment led to significantly greater DNA damage, as indicated by γH2AX[+] cells in the liver, compared with Sham+DEN at 4 months post-infection (Supplementary Fig. 3b, c). Likewise, HBV+DEN-treated liver showed elevated high-mobility group box 1 (HMGB1) expression, a well-known damage-associated molecular pattern (DAMP) molecule and TLR4 agonist[46–48], in hepatocytes compared with Sham+DEN-treated liver at 4 months post-infection (Fig. 2h, i and Supplementary Fig. 3d). HBV genome expression alone did not impact HMGB1 expression in the liver (Supplementary Fig. 3e). Serum HMGB1 levels were also elevated in HBV+DEN compared with HBV+PBS and Sham+PBS groups (Supplementary Fig. 3f), indicating that increased HMGB1 was secreted from the nucleus to function as a DAMP molecule. These findings indicate that the HMGB1/TLR4/TBK1/IRF3 axis activates IL-33 expression in HBV-positive liver that is exposed to carcinogens, which may lead to hepatitis and cancer development.

### IL-33 cytokine is required for HBV plus DEN-induced liver cancer development

To investigate the role of IL-33 in HBV+DEN-induced liver cancer, we subjected *Il33* knockout (Il33[KO]), ST2 knockout (ST2[KO]), and *Il33* transcription factor, *Irf3*, knockout (Irf3[KO]) mice to HBV+DEN liver carcinogenesis protocol (Fig. 1a). HBV+DEN-treated Il33[KO], ST2[KO], and Irf3[KO] mice showed significantly lower liver tumor burden and decreased PCNA[+] proliferating hepatocytes compared with HBV+DEN-treated WT mice at 8 months post-infection (Fig. 3a–d and Supplementary Fig. 4a). The comparable reduction in liver tumor burden observed in HBV+DEN-treated Il33[KO] and ST2[KO] mice demonstrated that IL-33's cytokine function, rather than its nuclear function[42], was responsible for HBV

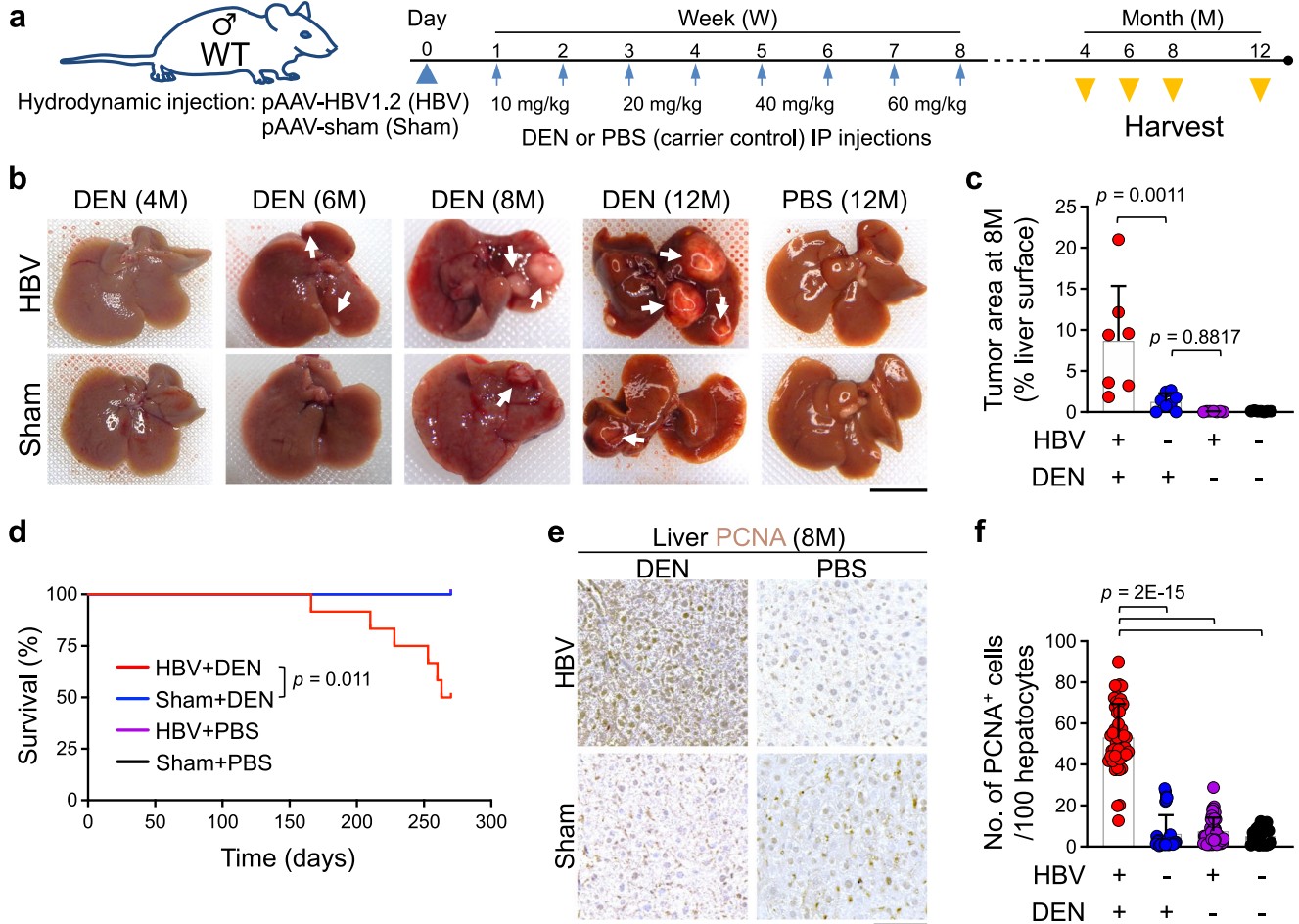

**Fig. 1 | HBV plus carcinogen promotes liver cancer development. a** The schematic diagram of liver carcinogenesis protocol. **b** Representative macroscopic images of the liver from WT mice that underwent liver carcinogenesis protocol at 4, 6, 8, and 12 months (M) post-infection. Arrows point to liver tumors. **c** Tumor burden measured as % liver surface area of WT mice that underwent liver carcinogenesis protocol at 8 months post-infection. $n = 7$ mice in HBV+DEN group, $n = 8$ mice in Sham+DEN group, $n = 7$ mice in HBV+PBS group, and $n = 7$ mice in Sham+PBS group. Each dot represents a mouse. Experimental data were verified in two independent experiments. **d** Survival of WT mice underwent liver carcinogenesis protocol. $n = 12$ mice in HBV+DEN group; $n = 10$ mice in Sham+DEN group, $n = 16$ mice in HBV+PBS group, and $n = 10$ mice in Sham+PBS group. Log-rank test.

**e** Representative images of PCNA stained liver from WT mice that underwent liver carcinogenesis protocol at 8 months post-infection. **f** Quantification of PCNA[+] hepatocytes in WT mice subjected to liver carcinogenesis protocol at 8 months post-infection. PCNA[+] cells per 100 hepatocytes were counted in five to eight randomly selected high power field (HPF) images of each liver. Each dot represents an HPF image. Liver samples were from $n = 6$ mice in HBV+DEN group, $n = 7$ mice in Sham+DEN group, $n = 7$ mice in HBV+PBS group, and $n = 4$ mice in Sham+PBS group. Graphs show mean + SD, (**c**, **f**) one-way ANOVA with Tukey's multiple comparison test, scale bars: 1 cm (liver images) and 100 μm (histology). Source data are provided as a Source Data file.

+DEN-mediated liver cancer development. Thus, we next examined which ST2-expressing immune cell type(s) is most suppressed in Il33[KO] compared with WT liver treated with HBV+DEN[49]. ST2[+] Tregs exhibited the most significant reduction compared with other immune cells in HBV+DEN-treated Il33[KO] versus WT liver at 4 months post-infection (Fig. 3e and Supplementary Fig. 4b). Tregs create a tumor-promoting environment by releasing transforming growth factor beta (TGF-β) and IL-10[50,51]. IL-2, a critical cytokine for maintaining Tregs[52–55], increased hepatic Tregs proliferation in combination with IL-33 (Fig. 3f). Interestingly, IL-33 addition to IL-2 significantly upregulated TGF-β1 and IL-10 expression by ST2[+] hepatic Tregs compared with IL-2 alone (Fig. 3g, h). Similarly, IL-33 addition to IL-2 significantly upregulated TGF-β1 and IL-10 expression by ST2[+] splenic Tregs (Supplementary Fig. 4c–e). Interestingly, we detected ST2[+] Tregs in HBV-associated chronic hepatitis in humans, which were increased compared with healthy control livers (Supplementary Fig. 5a)[56]. We found that ST2[+] Tregs were increased in HBV-associated immune active hepatitis compared with chronic resolved hepatitis and healthy controls (Supplementary Fig. 5b, c)[57]. These findings demonstrate that IL-33

supports ST2[+] Tregs in HBV+DEN-treated liver, which upregulate TGF-β and IL-10 expression in response to IL-33.

## IL-33/Treg axis mediates HBV plus DEN-induced liver cancer development

To determine whether IL-33-activated Tregs contributed to cancer development in HBV+DEN-treated mice, we subjected *Foxp3[Cre], ST2[flox/flox]* (TregST2[CKO]) mice in which ST2 was specifically deleted in Tregs to HBV+DEN liver carcinogenesis protocol (Fig. 1a). HBV+DEN-treated TregST2[CKO] mice showed significantly lower liver tumor burden, improved liver architecture, and decreased PCNA[+] proliferating hepatocytes compared with HBV+DEN-treated WT mice at 8 months post-infection (Fig. 4a–e and Supplementary Fig. 6a). A significantly fewer Tregs expressed IL-10 in TregST2[CKO] compared with WT liver (Fig. 4f and Supplementary Fig. 6b-d). Accordingly, loss of ST2 expression on Tregs compromised their immunosuppressive function, leading to increased CD3[+] T and CD8[+] T cell infiltration in the HBV+DEN-treated liver at 4 months post-infection (Fig. 4g-i). In addition, activated CD44[+]CD8[+] T cells were significantly increased in

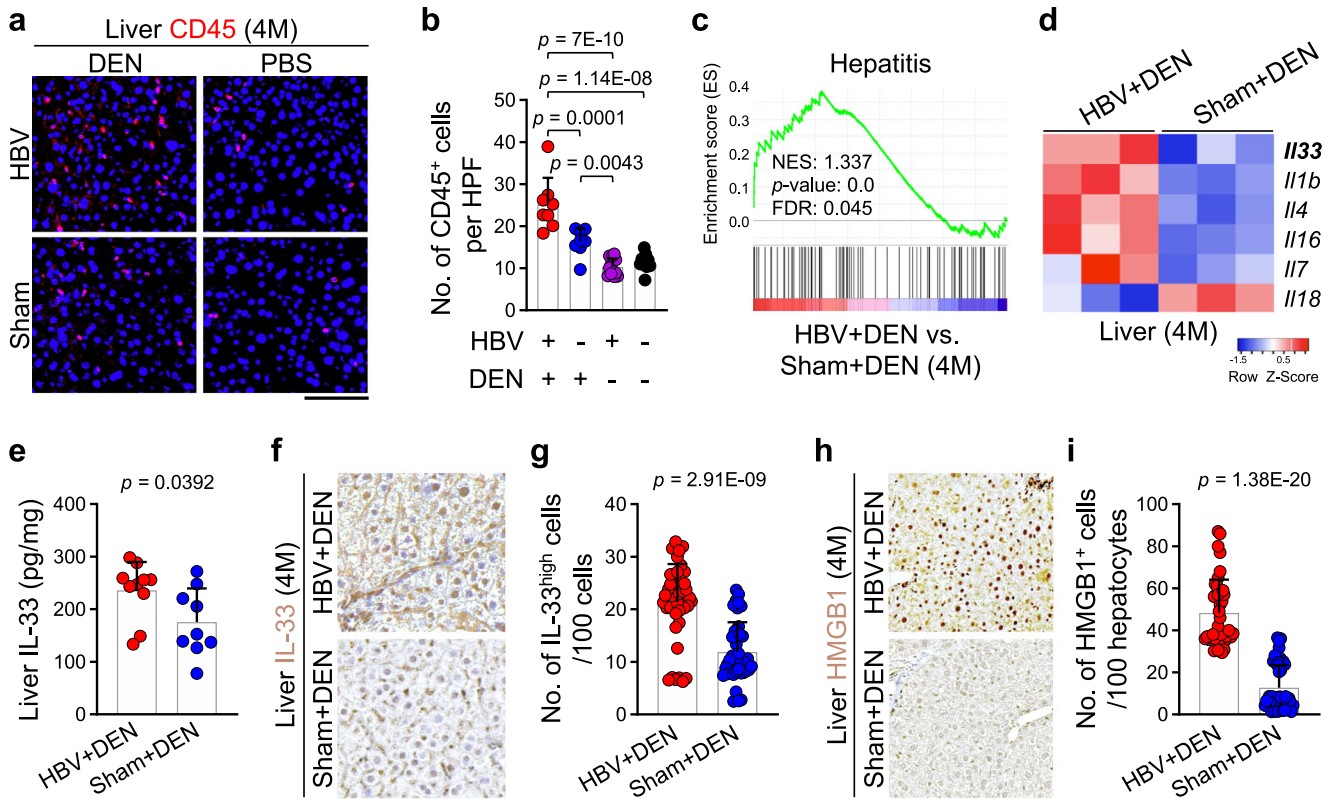

**Fig. 2 | HBV plus carcinogen-induced liver cancer development is preceded by chronic inflammation and IL-33 induction in the liver. a** Representative images of CD45 stained liver tissues treated with carcinogenesis protocol at 4 months post-infection. **b** Quantification of CD45[+] leukocytes in WT liver subjected to carcinogenesis protocol at 4 months post-infection. Each dot represents the average number of CD45[+] cells per 10 randomly selected HPF images from each liver. Liver samples were from $n = 8$ mice in HBV+DEN group, $n = 8$ mice in Sham+DEN group, $n = 12$ mice in HBV+PBS group, and $n = 10$ mice in Sham+PBS group. **c** The enrichment plot of hepatitis gene set from differentially expressed gene list of HBV+DEN-treated liver compared with Sham+DEN-treated liver at 4 months post-infection. $n = 3$ mice in each group, Kolmogorov-Smirnov test. **d** Heatmap of differentially expressed interleukin genes between HBV+DEN- and Sham+DEN-treated liver at 4 months post-infection. **e** IL-33 protein levels in HBV+DEN- ($n = 10$ mice) versus Sham+DEN-treated liver ($n = 9$ mice) at 4 months post-infection. Each dot represents a mouse. **f** Representative images of IL-33 stained HBV+DEN- and Sham+DEN-treated liver at 4 months post-infection. **g** Quantification of IL-33[high] cells per 100 cells in the HBV+DEN ($n = 8$ mice) compared with Sham+DEN-treated liver ($n = 6$ mice) at 4 months post-infection. IL-33[high] cells per 100 cells were counted in five to eight randomly selected HPF images per liver. Each dot represents an HPF image. **h** Representative images of HMGB1 stained HBV+DEN- and Sham+DEN-treated liver at 4 months post-infection. **i** Quantification of HMGB1[+] cells per 100 hepatocytes in the HBV+DEN ($n = 5$ mice) compared with Sham+DEN-treated liver ($n = 7$ mice) at 4 months post-infection. HMGB1[+] cells per 100 hepatocytes were counted in five to eight randomly selected HPF images per liver. Each dot represents an HPF image. Graphs show mean + SD, (**b**) one-way ANOVA with Tukey's multiple comparison test, (**e, g, i**) two-sided unpaired $t$-test, scale bar: 100 μm. Source data are provided as a Source Data file.

HBV+DEN-treated Treg-ST2[CKO] compared with WT liver (Supplementary Fig. 6e, f). Thus, the IL-33/Treg axis plays a critical role in mediating cancer development in the liver exposed to HBV plus carcinogens.

**Pitavastatin suppresses hepatitis and HCC**

Pitavastatin is a safe and widely used lipid-lowering medication that suppresses the TBK1-IRF3-IL-33 signaling pathway[39]. To determine the role of pitavastatin for liver cancer prevention in HBV-infected individuals, we treated WT mice with 2 mg/kg pitavastatin or PBS alone (carrier control) once a week during the HBV+DEN liver carcinogenesis protocol until harvest, which suppressed IL-33 levels (Supplementary Fig. 7a). Importantly, pitavastatin treatment significantly lowered liver tumor burden, improved liver architecture, and decreased PCNA[+] proliferating hepatocytes compared with carrier control in HBV+DEN-treated WT mice at 8 months post-infection (Fig. 5a–e and Supplementary Fig. 7b). To extend our findings to humans, we measured serum IL-33 levels in hepatitis patients and healthy individuals. HBV[+] and HBV[-] hepatitis patients showed similarly elevated serum ALT levels compared with healthy controls (Supplementary Fig. 7c). HBV-associated hepatitis patients showed significantly higher serum IL-33

levels compared with healthy controls, which were also higher than non-HBV hepatitis patients (Fig. 5f, g). Interestingly, serum IL-33 levels were higher in HBV[+] versus HBV[-] patients with other liver diseases, including HCC, cirrhosis, hepatic cyst, fatty liver, and liver hemangioma (Supplementary Fig. 7d). Moreover, serum IL-33 levels positively correlated with serum HBsAg levels across all HBV[+] patients in the study (Supplementary Fig. 7e and Supplementary Data 1). Based on these findings, we performed an epidemiological study to determine whether pitavastatin use was associated with a reduced risk of hepatitis and HCC. We compared matched cohorts of patients from the TriNetX diamond network, a global health network comprising electronic medical record-derived data from over 200 million patients across 92 healthcare organizations in North America and Europe (Supplementary Table 1). The risk of hepatitis was significantly reduced in patients treated with pitavastatin compared with those treated with ezetimibe, another cholesterol-lowering agent commonly used in the clinic that doesn't affect the mevalonate pathway (OR 0.727; 95% CI 0.685-0.771; $p < 0.0001$) (Fig. 5h). Furthermore, the risk of HCC associated with hepatitis was markedly decreased in the pitavastatin-treated group compared with the ezetimibe cohort (OR 0.577; 95% CI 0.506-0.657; $p < 0.0001$) (Fig. 5h). Lastly, *IL33* and *IL1RL1* (ST2) gene expression

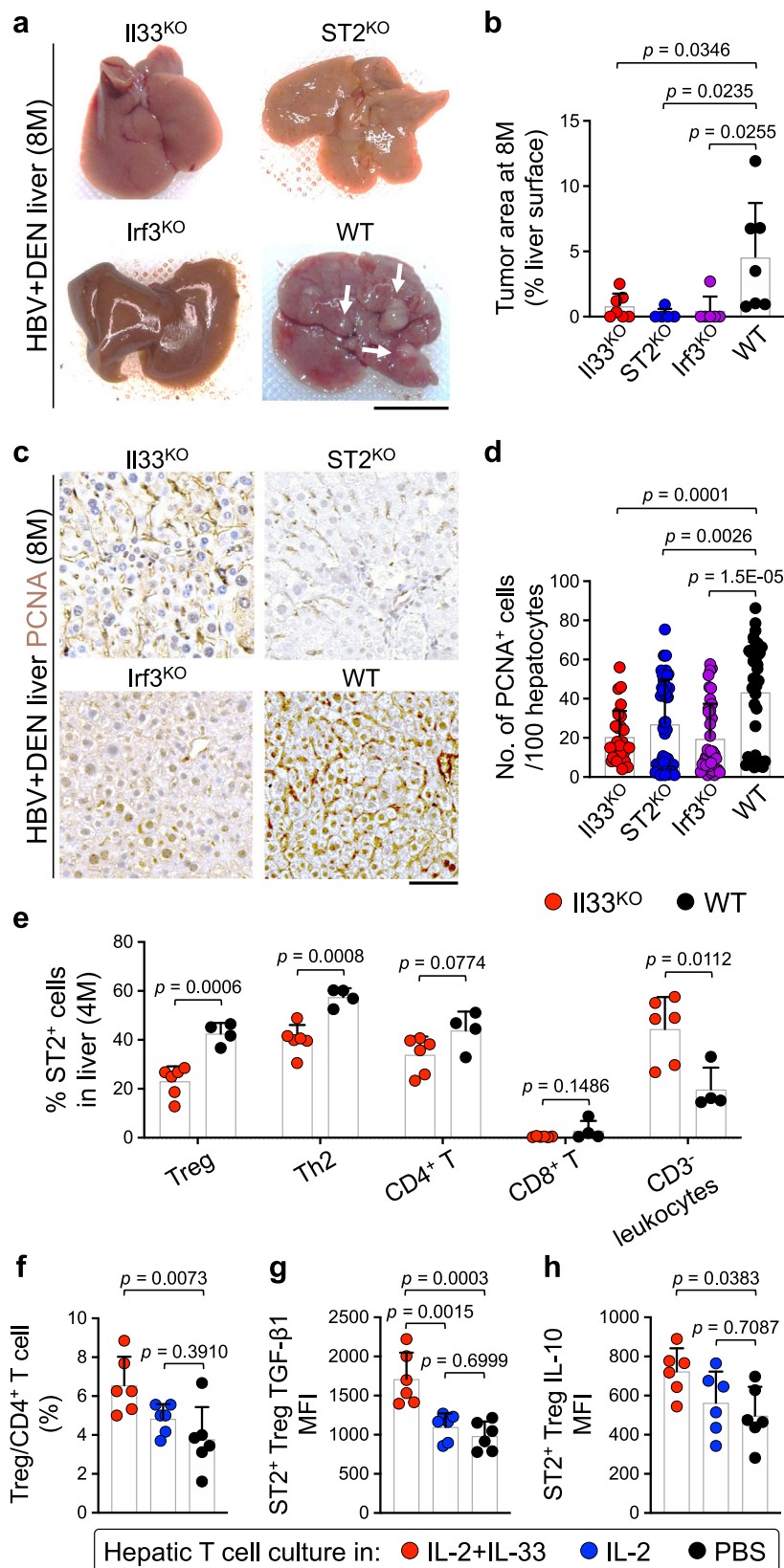

showed a negative correlation with CD8+ T cell and a positive correlation with Treg infiltration in human HCC samples from the Cancer Genome Atlas (TCGA) datasets (Fig. 5i, j). Collectively, HBV infection link to hepatitis and HCC is mediated by its impact on the immune response to environmental carcinogens, which are the primary drivers of hepatitis and HCC (Fig. 5k).

## Discussion

Our findings reveal the essential role of environmental carcinogens in linking HBV to hepatitis and HCC, highlighting the potential for effective strategies to prevent cancer development in HBV-infected individuals. HBV infection alone does not result in liver inflammation or cancer. Instead, HBV infection primarily exacerbates liver

**Fig. 3 | IL-33 cytokine signaling is required for HBV+DEN-induced liver cancer.**
**a** Representative macroscopic images of Il33[KO], ST2[KO], Irf3[KO], and WT liver treated with HBV+DEN at 8 months post-infection. Arrows point to liver tumors. **b** Tumor burden measured as % liver surface area of Il33[KO] (*n* = 7), ST2[KO] (*n* = 5), Irf3[KO] (*n* = 6), and WT mice (*n* = 7) that received HBV+DEN at 8 months post-infection. Each dot represents a mouse. Experimental data were verified in two independent experiments. **c** Representative images of PCNA stained Il33[KO], ST2[KO], Irf3[KO], and WT liver treated with HBV+DEN at 8 months post-infection. **d** Quantification of PCNA[+] hepatocytes in Il33[KO] (*n* = 4 mice), ST2[KO] (*n* = 7 mice), Irf3[KO] (*n* = 6 mice), and WT mice (*n* = 5 mice) that received HBV+DEN at 8 months post-infection. PCNA[+] cells per 100 hepatocytes were counted in five to eight randomly selected HPF images of

each liver. Each dot represents an HPF image. **e** Percent ST2[+] immune cell types in Il33[KO] (*n* = 6) and WT liver (*n* = 4) treated with HBV+DEN at 4 months post-infection. Each dot represents a mouse. **f** Hepatic Treg frequency as % total CD4[+] T cells in WT liver after incubation with IL-2 plus IL-33, IL-2, versus no cytokine treatment (PBS) control (*n* = 6). Each dot represents a mouse. **g** TGF-β1 mean fluorescence intensity (MFI) of IL-2 plus IL-33, IL-2, versus PBS-treated ST2[+] Tregs (*n* = 6). Each dot represents a mouse. **h** IL-10 MFI of IL-2 plus IL-33, IL-2, versus PBS-treated ST2[+] Tregs (*n* = 6). Each dot represents a mouse. Graphs show mean + SD, (**b**, **d**, **f**–**h**) one-way ANOVA with Tukey's multiple comparison test, (**e**) two-sided unpaired *t*-test, scale bars: 1 cm (liver images) and 100 μm (histology). Source data are provided as a Source Data file.

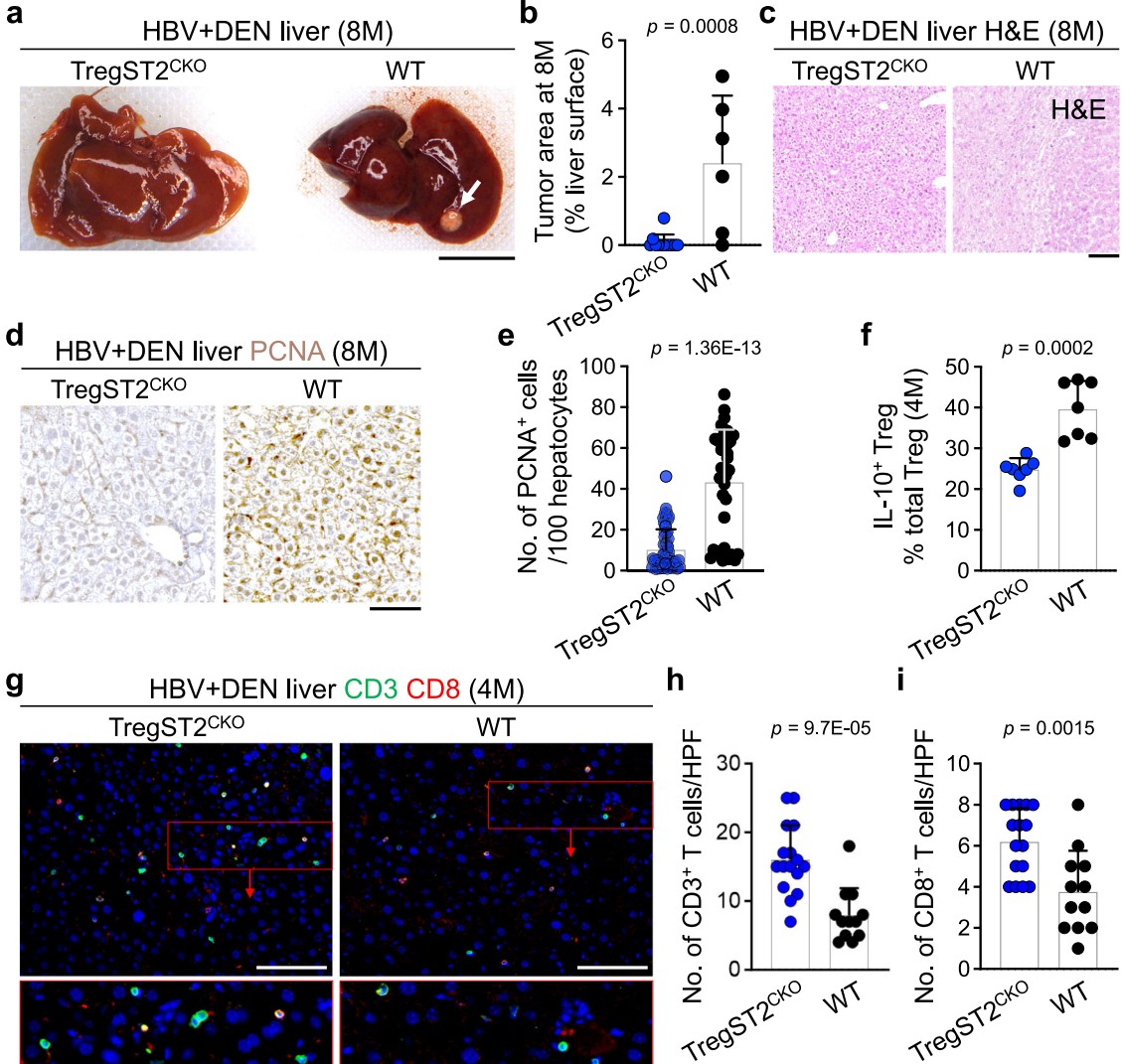

**Fig. 4 | IL-33/Treg axis is essential for HBV+DEN-induced liver cancer.**
**a** Representative macroscopic image of TregST2[CKO] and WT liver treated with HBV+DEN at 8 months post-infection. The arrow points to a liver tumor. **b** Tumor burden measured as % liver surface area of TregST2[CKO] (*n* = 12) and WT (*n* = 6) mice that received HBV+DEN at 8 months post-infection. Each dot represents a mouse. Experimental data were verified in two independent experiments. **c** Representative images of hematoxylin and eosin (H&E) stained TregST2[CKO] and WT liver treated with HBV+DEN at 8 months post-infection. **d** Representative images of PCNA stained TregST2[CKO] and WT liver treated with HBV+DEN at 8 months post-infection. **e** Quantification of PCNA[+] hepatocytes in TregST2[CKO] (*n* = 8) and WT (*n* = 5) mice that received HBV+DEN at 8 months post-infection. PCNA[+] cells per 100 hepatocytes were counted in five to eight randomly selected HPF images of each liver. Each dot represents an HPF image. **f** IL-10[+] Treg frequency in TregST2[CKO] (*n* = 7) and WT

(*n* = 7) liver treated with HBV+DEN at 4 months post-infection. Each dot represents a mouse. **g** Representative images of CD3 and CD8 stained TregST2[CKO] and WT liver treated with HBV+DEN at 4 months post-infection. Insets highlight T cells in the liver. **h** Quantification of CD3[+] T cells in TregST2[CKO] (*n* = 4) and WT (*n* = 3) liver treated with HBV+DEN at 4 months post-infection. CD3[+] cells were counted in three to four randomly selected HPF images per liver sample. Each dot represents an HPF image. **i** Quantification of CD8[+] T cells in TregST2[CKO] (*n* = 4) and WT (*n* = 3) liver treated with HBV+DEN at 4 months post-infection. CD8[+]CD3[+] cells were counted in three to four randomly selected HPF images per liver sample. Each dot represents an HPF image. Graphs show mean + SD, two-sided unpaired *t*-test, scale bars: 1 cm (liver images) and 100 μm (histology). Source data are provided as a Source Data file.

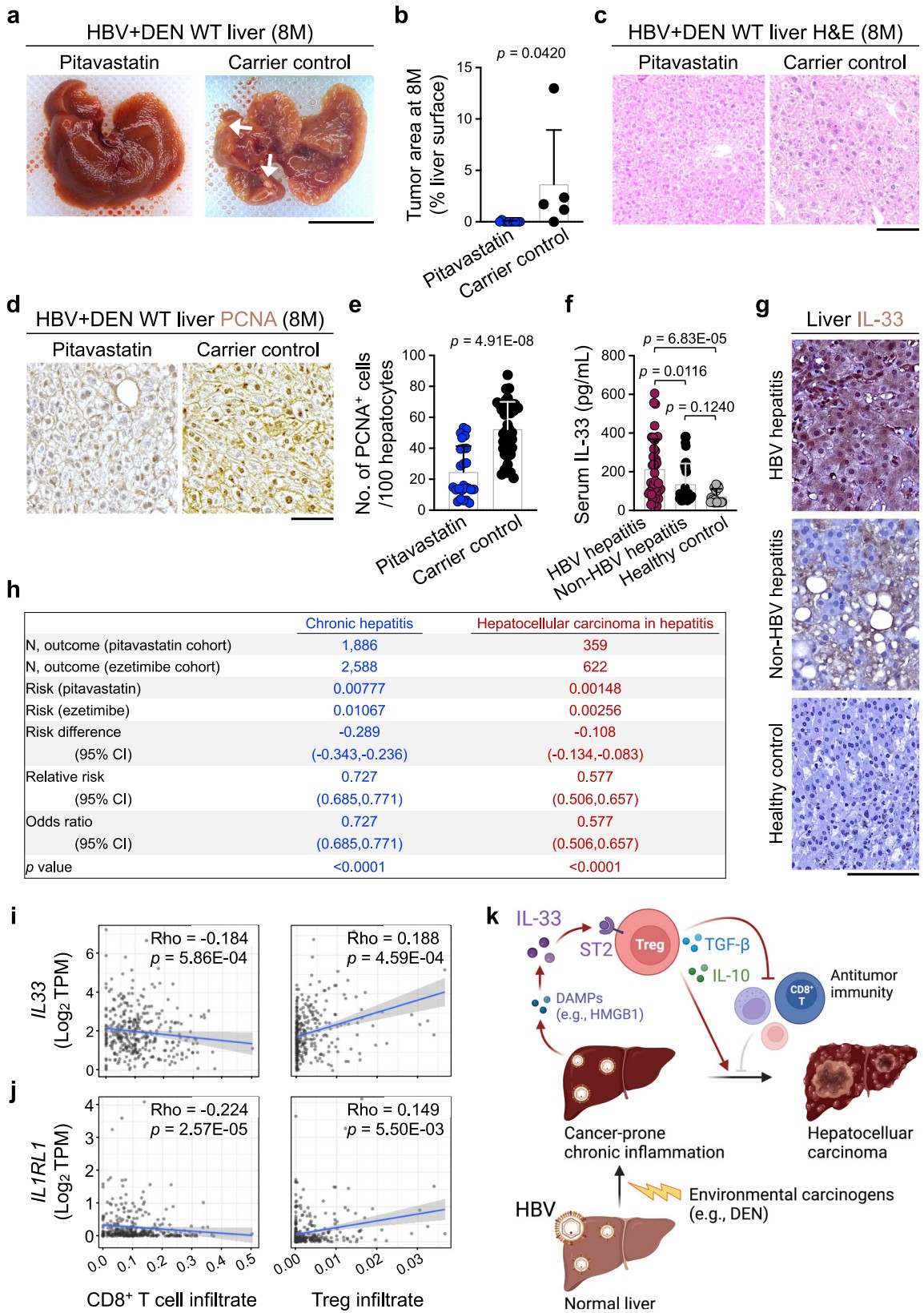

**h**

|  | Chronic hepatitis | Hepatocellular carcinoma in hepatitis |
|---|---|---|
| N, outcome (pitavastatin cohort) | 1,886 | 359 |
| N, outcome (ezetimibe cohort) | 2,588 | 622 |
| Risk (pitavastatin) | 0.00777 | 0.00148 |
| Risk (ezetimibe) | 0.01067 | 0.00256 |
| Risk difference | -0.289 | -0.108 |
| (95% CI) | (-0.343,-0.236) | (-0.134,-0.083) |
| Relative risk | 0.727 | 0.577 |
| (95% CI) | (0.685,0.771) | (0.506,0.657) |
| Odds ratio | 0.727 | 0.577 |
| (95% CI) | (0.685,0.771) | (0.506,0.657) |
| *p* value | <0.0001 | <0.0001 |

inflammation and early cancer development induced by carcinogens. This phenomenon is mediated by HBV, promoting DAMPs (e.g., HMGB1) expression in the HBV-containing hepatocytes that experience carcinogen-induced cellular damage. Importantly, we identify a key immune factor, IL-33, downstream of DAMP induction that significantly contributes to cancer development. IL-33 plays a pivotal role

in the pathogenesis of HBV plus carcinogen-induced liver injury by binding to ST2[+] Tregs in the liver. IL-33-activated Tregs produce higher levels of cytokines, TGF-β and IL-10, which promote chronic inflammation and early cancer development while inhibiting the recruitment of antitumor cytotoxic CD8[+] T cells[58–60]. Our research demonstrates that blocking the IL-33/Treg signaling axis, particularly through statin

**Fig. 5 | Pitavastatin blocks liver cancer development in chronic inflammation marked by elevated IL-33 expression. a** Representative macroscopic images of pitavastatin- and PBS-treated WT liver exposed to HBV+DEN at 8 months post-infection. Arrows point to liver tumors. **b** Tumor burden measured as % liver surface area of pitavastatin- ($n = 10$) and PBS-treated ($n = 5$) WT mice that received HBV+DEN at 8 months post-infection. Each dot represents a mouse. Experimental data were verified in two independent experiments. **c** Representative images of H&E stained pitavastatin- and PBS-treated WT liver exposed to HBV+DEN at 8 months post-infection. **d** Representative images of PCNA stained pitavastatin- and PBS-treated WT liver exposed to HBV+DEN at 8 months post-infection. **e** Quantification of PCNA[+] hepatocytes in pitavastatin- ($n = 4$) and PBS-treated ($n = 5$) WT mice that received HBV+DEN at 8 months post-infection. PCNA[+] cells per 100 hepatocytes were counted in five to eight randomly selected HPF images of each liver. Each dot represents an HPF image. **f** Serum IL-33 levels in HBV-positive hepatitis patients ($n = 40$), HBV-negative hepatitis patients ($n = 40$), and healthy controls ($n = 24$).

Each dot represents a patient serum sample. **g** Representative images of IL-33 stained liver tissues from HBV-positive hepatitis patients, HBV-negative hepatitis patients, and healthy control. **h** A retrospective cohort analysis of hepatitis and HCC in hepatitis risk in matched cohorts of patients treated with pitavastatin (test) versus ezetimibe (control) (two-sided two-proportion z-test). **i** The correlation between *IL33* expression levels and CD8[+] T cell and Treg cell infiltration in HCC samples ($n = 371$) from TCGA. **j** The correlation between *IL1RL1* expression levels and CD8[+] T cell and Treg cell infiltration in HCC samples ($n = 371$) from TCGA. **k** Schematic diagram outlining the mechanism that links HBV infection to increased hepatitis and HCC risk (Created in BioRender. Demehri, S. (2025) https://BioRender.com/taqks8w). Graphs show mean + SD, (**b**, **e**) two-sided unpaired *t*-test, (**f**) one-way ANOVA with Tukey's multiple comparison test, (**i**, **j**) two-sided *t*-test for Pearson correlation coefficient, scale bars: 1 cm (liver images) and 100 μm (histology). Source data are provided as a Source Data file.

treatment, can effectively suppress chronic hepatitis and prevent HCC development in HBV-infected individuals.

Despite the challenges associated with studying liver cancer development in the context of HBV infection, we have utilized an innovative HBV infection-like model employing AAV vector and hydrodynamic injection techniques[61,62] to gain insights into HBV-related immune responses and spontaneous liver cancer development. Chronic HBV infection is associated with a dysregulated immune response characterized by compromised adaptive immunity and increased liver damage[63,64]. Our research provides innovative insights into the mechanism linking HBV to liver damage, which is governed by exposure to carcinogens. We find that Tregs' role in HBV-associated liver pathogenesis is mediated by IL-33 induction in the liver[65]. IL-33 is an alarmin cytokine induced in response to cellular injury, which has pro- and anti-cancer effects in various contexts[43,66–70]. IL-33 induced by TLR3/4-TBK1-IRF3 signaling pathways drives cancer-prone chronic inflammation in the skin and pancreas[39,42,71]. However, IL-33 can enhance the activity of CD8[+] T cells and natural killer cells, thereby inhibiting cancer development[72,73]. We find that IL-33 induction is associated with HMGB1 upregulation in the liver, a known TLR4 ligand[46,74]. Interestingly, elevated HMGB1 expression is linked to various liver diseases, including hepatitis B, liver fibrosis, and liver cancer[75–78]. Previous research has shown that the HBV X protein could increase HMGB1 expression levels[78,79]. Furthermore, we find that the TBK1-IRF3 pathway is activated and essential for cancer development in the HBV plus carcinogen-treated liver. Pitavastatin inhibits the TBK1/IRF3 pathway by blocking TBK1 membrane recruitment, which is crucial to activating TBK1[39]. Thus, the TLR3/4-TBK1-IRF3-IL-33 signaling pathway activation by HBV plus carcinogen is a mechanism that explains hepatitis and HCC risk associated with HBV infection, which can be blocked by statin to suppress hepatitis and HCC.

Although IL-33 induction in the liver can serve to combat HBV infection[80], it is also elevated in other liver pathologies, including non-alcoholic steatohepatitis and intrahepatic cholangiocarcinoma[81–83]. In fact, IL-33 plays an important role in several liver diseases[84–87], including obesity-associated HCC[88]. IL-33 targets innate lymphoid cell (ILC) 2 and T helper 2 (Th2) cells, which contribute to liver fibrosis by inducing the expression of IL-13[85,87]. Moreover, the IL-33/ST2 axis promotes the proliferation of Tregs in the murine liver[89]. IL-33 supports liver cancer growth by remodeling the tumor microenvironment, leading to the secretion of tumor-promoting factors such as vascular endothelial growth factor[86]. Therefore, targeting IL-33 signaling pathways using statins has the potential to inhibit liver fibrosis and inhibit HCC development. Our epidemiological data support the use of pitavastatin as an effective therapy for hepatitis and HCC. By inhibiting the mevalonate pathway, statins not only block IL-33 induction but also disrupt viral replication by controlling the expression of adhesion molecules on the surface of host cells[90–93], potentially extending their therapeutic benefits to controlling HBV infection.

There are several mouse models that mimic hepatitis B virus infection, including transgenic mouse models and adenovirus-based models[94–96]. Although our hydrodynamic injection model does not involve live HBV infection and may not fully represent HBV infection in humans, it leads to the persistence of the HBV genome and its gene expression in the liver following the hydrodynamic injection. To model liver cancer development in mice, DEN injection into 14-day-old male mice has been widely used[97,98]. However, to specifically investigate HBV-associated liver cancer development, we established a protocol that incorporates HBV infection in young animals followed by several DEN injections to enable cancer development in adult mice. Although RNA sequencing suggested the elevation of other cytokines (e.g., IL-β and IL-4) in HBV+DEN-treated liver, we did not observe any significant elevation for these cytokines at the protein level. Interestingly, *Il18* gene expression was found to be suppressed in the HBV+DEN-treated liver, which warrants future investigations to determine its impact on hepatitis and HCC development.

In conclusion, our research sheds light on the intricate interplay between HBV infection and carcinogen exposure to generate immune dysregulation in the liver that leads to cancer development. By targeting the IL-33/Treg axis with pitavastatin, we offer a promising approach for the prevention and treatment of chronic hepatitis and its cancer sequelae in HBV-infected individuals. We propose the potential of combining statin therapy with conventional HBV treatments like entecavir to improve patient outcomes[39].

## Methods

### Study approval
Massachusetts General Hospital IACUC approved the animal studies. The First Affiliated Hospital of USTC Ethics Committees approved the human sample studies.

### Human samples
All serum and liver biopsy samples were collected from the First Affiliated Hospital of University of Science and Technology of China (Hefei, China). All patients in this study provided written informed consent for sample collection and data analysis. The healthy control liver biopsy came from the distal healthy liver tissue of surgical resection samples from patients with liver hemangioma. HBV[+] is defined as HBsAg> 0.08 IU/mL. This study was approved by the Ethics Committees of the First Affiliated Hospital of University of Science and Technology of China (USTC) (2020-KYLS-124). The clinical data used in this study are anonymized and aggregated (Supplementary Data 1).

### Animal studies
All mice were housed under pathogen-free conditions in an animal facility at Massachusetts General Hospital following animal care regulations. Mice were housed on a 12 light /12 dark cycle, at -18–23 °C with 40%–60% humidity. Irf3[KO] mice were purchased from the Riken

BioResource Research Center (Ibaraki, Japan). Il33[KO] mice were a gift from Dr. Marco Colonna, and ST2[KO] mice were a gift from Dr. Peter Nigrovic. TregST2[CKO] mice (Foxp3Cre RRID: IMSR_JAX: 016959; ST2[flox/flox]) were generated by Dr. Richard T. Lee and Dr. Diane Mathis. All mutant mice were maintained on the C57BL/6 background. WT C57BL/6 mice were purchased from the Jackson Laboratory (Bar Harbor, ME). Massachusetts General Hospital IACUC permitted a 20 mm maximal tumor diameter, and the maximal tumor size was not exceeded. Mice were euthanized under anesthesia at predetermined time points, including tumors reaching 20 mm diameter, weigh loss greater than 20% of the initial total body weight, a weigh increase of 20% due to ascitic fluid, a hunchbacked appearance, or a moribund state.

### Hydrodynamic injection-based mouse HBV infection-like model
The pAAV-HBV1.2 plasmid, which contains a 1.2 fold over-length HBV genotype A genome, was kindly provided by Dr. Per-Jer Chen. 6 µg of pAAV-HBV1.2 plasmid DNA was hydrodynamically injected into the tail veins of 4-week-old male mice. An empty vector (pAAV-sham) is used as a control. Rapidly injecting a large volume of plasmid (10% body weight in 5–7 sec) results in HBV DNA transfection mostly into hepatocytes[62] and provides an immunocompetent mouse model for chronic HBV infection[61]. HBsAg serum levels were determined by ELISA once a month after pAAV-HBV1.2 plasmid injection.

### DEN-induced HCC model
To establish an HBV-associated hepatocellular carcinoma model, male C57BL/6 mice received DEN (Diethylnitrosamine, Merck, Darmstadt, Germany, catalog no. 73861) administration within 1 week of pAAV-HBV1.2/empty vector plasmid hydrodynamic injection. DEN was intraperitoneally administrated for eight consecutive weeks at increasing concentrations:10 mg/kg body weight for the first two weeks, 20 mg/kg body weight for the second two weeks, 40 mg/kg body weight for the third two weeks, and 60 mg/kg body weight for the last two weeks. Liver tissues were harvested at different time points following both the HBV challenge and DEN exposure.

### Pitavastatin treatment
Pitavastatin in dimethyl sulfoxide (DMSO) (5 mg/mL) was made as a stock solution for the mice experiment. Mice were treated intraperitoneally with 2 mg/kg pitavastatin (Selleck Chemicals LLC, Houston, TX, catalog no. S1759) in 100 µL PBS or the same volume of DMSO with PBS. Pitavastatin was given once a week until harvest.

### Western blot
Liver tissues were meshed and lysed by 0.1% TWEEN-20 (Merck, catalog no. P1379) in PBS. Tissue lysates were frozen in liquid nitrogen and thawed by incubation at 37 °C for further lysis. Lysates were sonicated for 10 seconds and centrifuged at high speed (15,000 g). After checking the protein concentration with Pierce BCA protein assay kit (Thermo Fisher Scientific, Waltham, MA, catalog no. 23225) in each sample, the identical amounts of total proteins were loaded onto Mini-PROTEIN TGX™ Gels (BIO-RAD, Hercules, CA, catalog no.456-1083 or 456-1086) with 1X Tris/Glycine/SDS buffer (BIO-RAD, catalog no.1610732). After 40 min (according to protein size, 200 Voltage), samples were transferred to Immobilon−P membrane (Merck, catalog no. IPVH00010) with Transfer buffer (Boston Bioproducts, Ashland, MA, catalog no. BP-190). For blocking step, the membranes were incubated with 3% bovine serum albumin (Thermo Fisher Scientific, catalog no. BP1600) or 5% Skim milk (BD biosciences, San Jose, CA, catalog no. 232100) in 1X Tris-Buffered Saline (Boston Bioproducts, Milford, MA, catalog no. BM301X) containing 0.1% TWEEN, called TBS-T for 30 min. After washing with TBS-T three times, the membranes were subjected to immunoblot with proper antibodies overnight at 4 °C. The following day, membranes were incubated with appropriate rabbit secondary antibody after washing. Membranes were developed with Pierce ECL Western blotting substrate kit (Thermo Fisher Scientific, catalog no. 32106) and signals were confirmed with BIO-RAD (BIO-RAD, catalog no. 1700140). Primary and secondary antibodies are listed in Supplementary Data 2.

### Histology, immunohistochemistry, and Immunofluorescence
Tissue samples were collected and fixed in 4% paraformaldehyde (Merck, catalog no. P6148) overnight at 4 °C. Next, tissues were dehydrated in PBS and 30, 50, 70% ethanol, processed, and embedded in paraffin. Five to seven µm sections of paraffin-embedded tissues were placed on slides, deparaffinized, and stained with Hematoxylin & Eosin (H&E). For immunohistochemistry (IHC), antigen retrieval was performed in 500 µL of antigen unmasking solution (VECTOR Laboratories, catalog no. H3300) diluted in 50 mL distilled water using a Cuisinart pressure cooker for 20 min at high pressure. Slides were washed three times for three minutes each in 1X TBS with 0.025% Triton X-100. Sections were blocked with 3% bovine serum albumin (Thermo Fisher Scientific, catalog no. BP1600) and 5% goat serum (Merck, catalog no. G9023) for 1 h. Slides were incubated overnight at 4 °C with a primary antibody diluted in TBS containing 3% BSA (Supplementary Data 2). The next day, slides were washed as above and incubated in 100 µL biotinylated secondary antibody (VECTOR Laboratories, catalog no. PK-6200) for 30 min. Slides were subjected to a 100 µL mixture of reagents A and B from VECTASTAIN Elite ABC universal kit Peroxidase (VECTOR Laboratories, catalog no. PK-6200) for 30 min. After washing, slides were incubated with 100 µL ImmPACT DAB chromogen staining (VECTOR Laboratories, catalog no. SK-4105) for a few minutes (depending on the signal, usually less than 1–2 min). Finally, slides were dehydrated in 70, 90, 100% ethanol and xylene and mounted with a coverslip using three drops of mounting media. For immunofluorescence (IF) staining, rehydrated tissue sections were permeated with 1X PBS supplemented with 0.2% Triton X-100 for 5 min. Antigen retrieval was used similarly to IHC. Slides were washed three times for 3 min each in 1X PBS with 0.1% Tween-20. For the blocking step, slides were blocked with 3% bovine serum albumin and 5% goat serum for 1 h. The slides were incubated overnight at 4 °C with primary antibodies. The next day, slides were washed as above and stained for 2 h at room temperature with secondary antibodies conjugated to fluorochromes. Next, slides were incubated with 1:5000 DAPI (Thermo Fisher Scientific, catalog no. D3571) with PBS for 3 min at room temperature, then washed as above. Slides were covered with Prolong Gold Antifade Reagent (Thermo Fisher Scientific, catalog no. P36930). The number of positive cells was counted in randomly selected high-power field (HPF, 200 x magnification) images in a blinded manner by a trained investigator. A pathologist reviewed clinical samples.

### RNA-sequencing
Total RNA was extracted from the mouse liver samples harvested at 4 months after HBV infection, using the RNeasy Mini Kit (Qiagen, Hilden, Germany, catalog no. 74104) following the manufacturer's instructions. For each sample, 1 µg of RNA was retrotranscribed into cDNA using SuperScript III Reverse Transcriptase (Thermo Fisher Scientific, catalog no.18080093) as indicated by the provider. Novogene performed and analyzed RNA sequencing (RNA-seq). Samples were barcoded and run on a HiSeq 2500 in 50 bp/50 bp paired-end run using HiSeq3000 SBS Kit (Illumina) for 30-40 million paired reads. The raw sequencing FASTQ files were aligned against mm10 assembly by STAR (v2.6.1b). FPKM (Fragments Per Kilobase Million) values were then computed from gene level counts by using fpkm function from the R package "DESeq2". Original data are available in the NCBI Gene Expression Omnibus (GEO) with accession number GSE269528.

## Single-cell RNA-sequencing

The normal liver and chronic hepatitis B single-cell datasets were downloaded from the GEO database. The single-cell RNA-sequencing (scRNA-seq) data of normal human liver tissues ($n = 3$) and chronic hepatitis B tissues ($n = 13$) were from GSE234241[56]. The single-cell RNA sequencing data of immune cells from HBV-associated immune active (IA), chronic resolved (CR), and HBV-free healthy controls (HC) were from GSE182159[57]. After filtering out cells with a mitochondrial gene ratio of less than 25%, Seurat (v5.1.0) was used for data analysis. Principal component analysis (PCA) was performed on single-cell samples, and the top 20 principal components (PC) were selected for subsequent analysis. The uniform manifold approximation and projection (UMAP) was used to perform an overall dimensionality reduction analysis on the top 20 PC pairs of samples. Data visualization was conducted using the ggplot2 package (v3.5.2).

## TCGA data analysis

The correlation of *IL33* and *IL1RL1* expression with CD8+ T cell and Treg infiltration in HCC samples included in the TCGA dataset was obtained from the "GENE" module in the Tumor Immune Estimation Resource (TIMER) (https://cistrome.shinyapps.io/timer/) database[99].

## PCR

DNA was extracted from mouse tissue using KAPA Express Extract buffer, and KAPA Express Extract enzyme from Kapa Genotyping kit (Kapa Biosystems Inc, Wilmington, MA, catalog no. KK7302). After the tissue lysis step, PCR was performed using 2X KAPA2G Fast genotyping mix from the Genotyping kit and the primers that are listed in Supplementary Data 3.

## Epidemiological analysis

Analysis was performed using de-identified data from the TriNetX Diamond Network. A search query was used to identify the cohort of patients within the network who had received pitavastatin. Eligible patients were identified based on the presence of corresponding RxNorm concept unique identifiers (RXCUI) in the patient's electronic medical records. Using International Classification of Diseases Tenth Revision (ICD-10) codes, patients with a history of chronic hepatitis and HCC prior to statin initiation were excluded from the cohorts to reduce confounding. The control cohort for each analysis included all patients within the network who had received ezetimibe but had no recorded statin use and patients with a history of any of the diagnoses mentioned above before ezetimibe initiation were also excluded. The index event for all analyses was the initiation of pitavastatin for test cohorts and the initiation of ezetimibe for the control cohort. Cases and controls were matched using 1:1 propensity score-matching for age at index event, sex, race, and ethnicity using "greedy nearest neighbor matching" and a caliper of 0.1 pooled standard deviations. Baseline characteristics were reported by count and percentage of the total for categorical variables and mean and standard deviations (SD) for continuous variables. Relative risks are presented with 95% confidence intervals. *P* values are uncorrected and based on Z-tests or Fisher's exact tests. Statistical analyses were performed in real-time using the TriNetX platform.

## HBV detection

HBsAg in the mouse serum was quantified with Quicktiter™ HBsAg ELISA kit (Cell Biolabs, San Diego, CA, catalog no. VPK-5004) as per the supplier's instructions. HBsAg in human serum was determined by chemiluminescent microparticle immunoassay with anti-HBs Reagent Kit (Abbott, Chicago, IL, catalog no. K4682C). HBV DNA values were measured in human serum using quantitative real-time PCR HBV DNA assay (DA-AN Gene, China, catalog no. DA0041). To detect HBV DNA in mouse serum, we used HBV TaqMan PCR kit (NORGEN BIOTEK,

Canada, Catalog no. TM29250), which measures relative HBV concentration in comparison to the positive control provided in the kit.

## Serum alanine aminotransferase activity assay

Serum alanine aminotransferase (ALT) activity was measured for mice and patients with hepatitis and other liver diseases. ALT activity was determined by Alanine Aminotransferase Activity Assay Kit (Sigma-Aldrich, St. Louis, MO, catalog no. MAK052). For the colorimetric assay, the absorbance at 570 nm was determined and proportional to the pyruvate generated as the ALT activity. ALT activity is reported as nmole/min/mL = milliunit/mL, where one milliunit (mU) of ALT is defined as the amount of enzyme that generates 1.0 nmole of pyruvate per minute at 37 °C.

## Flow cytometric analysis

Livers from mice were harvested, mashed through a 100 μm filter and centrifuged at 55 g for 1 min to remove hepatocytes. The samples were then centrifuged at 3000 g to collect cells. The pellets were resuspended in 40% isotonic Percoll (Merck, Catalog no. GE17-0891-01), a mixture of Percoll with 1X PBS and 10X PBS, and the samples were gently layered onto 70% Percoll. After centrifugation at 2000 g with very low deceleration conditions, cells were collected from the interface between the 40% and 70% Percoll layers. The cells were washed with PBS and incubated in RPMI 1640 (Life Technologies, Carlsbad, CA, catalog no. 21870076) supplemented with 10% FBS (Corning, Manassas, VA, USA) and 1% penicillin and streptomycin (Thermo Fisher Scientific, catalog no. 14140122) (R10 media) along with a cell activator cocktail (Biolegend, San Diego, CA, catalog no. 423301) at 37 °C. After one hour of incubation, monensin (Biolegend, catalog no. 420701) and brefeldin A (Biolegend, catalog no. 420601) were added, and the cells were incubated for an additional three hours at 37 °C. The cells were stained with the following surface marker antibodies: anti-CD45, anti-CD3, anti-CD4, anti-CD8, anti-ST2 and anti-CD25 antibodies (Supplementary Data 2). They were then fixed and permeabilized using the True-Nuclear Transcription Factor Buffer Set (Biolegend, catalog no. 424401) for staining with anti-FoxP3, anti-GATA3, anti-IL-10, and anti-TGF-β antibodies (Supplementary Data 2). The stained cells were analyzed using a LSR Fortessa X-20 flow cytometer (BD Bioscience) and the data were processed with FlowJo software (Tree Star).

## ELISA

Commercial ELISA kits were used according to manufacturers' protocols. ELISA kit for the measurement of HMGB1 was purchased from Thermo Scientific (catalog no. EEL102). The mouse IL-33, IL-4, and IL-1β quantification kits were purchased from Biolegend (catalog no. 436407, 431104, 432604). The mouse IL-16 quantification kit was purchased from RayBiotech (catalog no. ELM-IL16). The human IL-33 quantification kit was purchased from MultiSciences (catalog no. 70-EK133-96).

## In vitro experiments

Livers and spleens from C57BL/6 mice were harvested and meshed through a 70 μm filter, and the cells were resuspended in RBC lysis buffer for 3 min. After washing with PBS, the CD4+ T cells were isolated using MojoSort mouse CD4+ T cells isolation kits (Biolegend, catalog no. 480006). After cell counting, $5 \times 10^6$ cells were resuspended in 1 mL of R10 media containing 2 μg/mL of CD28 antibody (BioXcell, Lebanon, NH, catalog no. BE0015-1). The cells were then plated in a 12-well plate coated with 10 μg/mL of CD3 antibody (BioXcell, catalog no. BE0001-1) and incubated at 37 °C. After 24 h, PBS, IL-2 (20 ng/mL), or IL-2 (20 ng/mL) plus IL-33 (200 ng/mL) was added to the cells. The next day, monensin and brefeldin A were added and the samples were incubated for an additional four hours at 37 °C. Finally, the samples were stained with the appropriate fluorochrome-conjugated antibodies for flow cytometry analysis.

## Statistical analysis and reproducibility

A two-sided *t*-test for the Pearson correlation coefficient was used for the correlation between *IL33* and *IL1RL1* expressions and immune cell infiltration, and the correlation between serum IL-33 and HBsAg levels in patients. Statistical differences between the three groups were analyzed using one-way ANOVA. Tukey's multiple comparison test was used to examine the differences in the mean ranks among all three possible pairwise comparisons. A two-sided unpaired *t*-test was used to test the significance of tumor area, PCNA-positive cells, leukocyte counts, protein expression levels, and other quantitative measurements. Comparisons of survival were performed with the log-rank test. The risk ratios of epidemiological data were compared using a two-tailed, two-proportion z-test. *p*-value < 0.05 is considered significant. Bar graphs show mean + SD. Sample sizes were not predetermined based on statistical methods but were chosen according to the standards of the field (at least three independent biological replicates for each condition).

## Reporting summary

Further information on research design is available in the Nature Portfolio Reporting Summary linked to this article.

## Data availability

All data needed to evaluate the conclusions in the paper are present in the paper and the Supplementary Information. RNA sequencing data can be accessed from NCBI database, GEO accession no.: GSE269528. The single-cell RNA sequencing data can be accessed from NCBI database, GEO accession no.: GSE234241 and GSE182159. Source data supporting the findings of this study are provided with this paper and is available in Figshare at https://doi.org/10.6084/m9.figshare.29270633.

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

## Acknowledgements

We thank Dr. Marco Colonna for the Il33^KO mice, which were generated with support from the Mucosal Immunology Studies Team (MIST) (U01; RFA-AI-15-023). We thank Dr. Peter Nigrovic for ST2^KO mice, and Dr. Richard T. Lee and Dr. Diane Mathis for TregST2^CKO mice. We thank Dr. Per-Jer Chen for the pAAV-HBV1.2 plasmid. SD holds a Career Award for Medical Scientists from the Burroughs Wellcome Fund. A.B. and J.H.P. were supported by the National Research Foundation of Korea grant funded by the Korea government (MSIT) (RS-2024-00343783). M.H. was supported by the Natural Science Foundation of Anhui, China (2008085MH253) and the National Natural Science Foundation of China (82473312). D.W. was supported by the National Natural Science Foundation of China (82370217) and the Natural Science Foundation of Anhui, China (2308085Y46). X.Z. was supported by the First Affiliated Hospital of Sun Yat-sen University. J.H.P., M.H., D.W., X.Z., M.M., M.A., and S.D. were supported by grants from the Burroughs Wellcome Fund and NIH (R01CA283214).

## Author contributions

M.H. and S.D. conceived the study. M.H., D.W., J.P.H., and S.D. designed the experiments. M.H., D.W., J.H., A.B., Y.X., X.Z., M.M., M.A., and J.H.P. performed the experiments and analyzed the data. M.R.C. and Y.R.S. performed the epidemiological study. M.H., D.W., J.H.P., and S.D. interpreted the data and wrote the manuscript.

## Competing interests

J.H.P. and S.D. are coinventors on a filed patent for the use of IL-33 inhibition in the treatment of cancer, fibrosis, and inflammation (PCT/US21/40725). Other authors state no conflict of interest.
