## [Transparent Peer Review file · Nature Communications]

Hepatitis B virus promotes liver cancer by modulating the immune response to environmental carcinogens

Corresponding Author: Dr Shadmehr Demehri

Version 0:

Reviewer comments:

Reviewer #1

(Remarks to the Author)
MS No. NCOMMS-24-63750

Hepatitis B virus promotes liver cancer by modulating the immune response to environmental carcinogens
by Jong Ho Park et al.

In this study, the authors demonstrated that HBV infection alone does not directly cause liver inflammation or cancer. Instead, they found that HBV alters the chronic inflammation induced by chemical carcinogens, such as DEN, to promote liver carcinogenesis. The combined administration of HBV and DEN activated the IL-33/regulatory T cell axis, which is essential for liver cancer development. Pitavastatin, used as an IL-33 inhibitor, successfully suppressed liver cancer induced by the HBV and DEN combination. The findings highlight the significant role of environmental carcinogens in increasing the risk of HBV-related hepatocellular carcinoma (HCC).

The authors propose that the combination of HBV infection and chemical carcinogens is crucial for driving chronic inflammation. However, it is well established that viral infections typically activate type I interferon (IFN) signaling, which contributes to inflammatory microenvironments. Therefore, it is recommended that the authors assess type I IFN signaling in their model. Additionally, many of the findings are derived from a mouse model, raising concerns about the clinical relevance to human HBV-associated HCC, which needs further investigation.

Major Concerns:

1. The authors identified cytokines such as IL-33, IL-1 β , IL-4, IL-16, and IL-7 as differentially expressed in AAV-infected, HBV-expressing hepatocytes. However, these cytokines are commonly expressed by various immune cells, not hepatocytes. Given the possible influence of hydrodynamic injection on cells around the liver sinusoid, including hepatic stellate cells, it is critical to confirm the specific cell types expressing these cytokines. IL-33, for instance, is known to be produced by hepatic stellate cells to suppress anti-tumor immunity (PMID: 35749514). The authors should use single-cell analysis or immunohistochemistry with cell-type-specific markers to verify the sources of these cytokines. Additionally, they should clarify why IL-18 expression was repressed (Figure 1j), considering that IL-1 β and IL-18 are activated via inflammasomes.
2. While the authors show that hepatocyte-derived IL-33 stimulates ST2-positive Treg cells, it is important to determine whether this occurs in human HBV infections as well. The authors should assess the prevalence of ST2-positive Treg cells in human HBV-infected liver tissues, as the mouse model may not fully capture the human immune environment.
3. The authors propose that HBV infection, combined with carcinogen exposure, leads to hepatocyte damage and DAMP release, triggering IL-33 expression. This hypothesis is intriguing but requires further investigation. The authors should specify which cell types express IL-33 upon DAMP stimulation. Moreover, given the variety of carcinogens in natural environments, what is the common molecular mechanism driving the second stimulation? Is it linked to DNA damage, necrotic cell death, or another process? Clarifying the immunological consequences of this chronic inflammation is crucial for understanding these findings.

4. The serum IL-33 levels in hepatitis patients (n=12, Figure 4f) are significantly elevated regardless of HBV infection. However, the sample size is too small to draw definitive conclusions about differences between HBV-positive and HBV-negative HCC patients. The authors should increase the sample size to strengthen their findings. Additionally, they should investigate the mechanism behind elevated IL-33 levels in non-HBV patients, as this could provide insight into the mechanism of IL-33 induction in liver diseases.

5. The use of pitavastatin as an inhibitor of IL-33-mediated pathways raises concerns. Pitavastatin is primarily an inhibitor of HMG-CoA reductase, which reduces cholesterol synthesis. The authors need to clarify how inhibiting HMG-CoA reductase impacts IL-33 expression. The reduced incidence of HCC might be related to lowered cholesterol levels, rather than direct inhibition of IL-33. Further mechanistic studies are required to bridge the gap between cholesterol metabolism and the IL-33 signaling pathway.

Reviewer #2

(Remarks to the Author)

Title: Hepatitis B virus promotes liver cancer by modulating the immune response to environmental carcinogens.

Summary:

In this study, the authors detail an investigation of the link between chronic hepatitis B virus (HBV) genome delivery to hepatocytes and the development of hepatocellular carcinoma (HCC). The HBV genome was delivered to hepatocytes via hydrodynamic injection of pAAV-HBV1.2, which alone was not sufficient for HCC development. It was determined that pAAV-HBV1.2 delivery combined with carcinogen administration (diethyl nitrosamine: DEN) increased HCC incidence and progression and as a result reduced survival compared to empty vector AAV mice treated with DEN. In mice deficient in ST2, IL-33, or IRF3, HBV+DEN treatment resulted in a significant reduction in HCC development. The progression of HCC was dependent on expansion of ST2+ Tregs via IL-33 production which was enhanced in HBV infected, DEN-treated mice. Inhibiting signalling through ST2 via Pitavastatin treatment resulted in reduced HCC development and improved survival. The results of this manuscript address an important knowledge gap regarding the development of HCC in patients with chronic HBV.

There are some major points for improvement of this manuscript including further contextualisation within the current literature, appropriate discussion of the limitations of the study, increased clarity regarding repetition of experiments, and improvement of flow cytometry data. These major concerns should be addressed to improve the scientific rigor and quality of this study.

Key results:

- Authors show that chronic HBV infection changes the response to chemical carcinogens, as in their model, HBV genome expression in hepatocytes exacerbated the development of HCC.
- The increased HCC development in AAV-HBV-infected mice was shown to be dependent on ST2 expression by Treg cells.
- Inhibition of ST2-IL-33 axis results in reduced liver tumour burden in mice with the HBV genome in hepatocytes treated with a carcinogen.

Significance & validity:

Authors have identified a therapeutic strategy for patients with HBV and HCC. Studies focused on HCC in isolation have previously identified an important role in IL-33 signalling in the progression of disease potentially limiting the significance of this part of the story. However, this study expands on the current knowledge by addressing the link between HBV-genome expressing hepatocytes and HCC. The authors demonstrate reduced tumour burden upon inhibition of IL-33-ST2 signalling axis in their model.

Major comments

Clarity and context

The Introduction requires careful rephrasing

1. Lines 49-53 could be re-phrased to more clearly state the current challenges in ensuring HBV immunity, it would be more precise to state a combination of factors contribute to the global threat of HBV:

Although prophylactic vaccination is available, a combination of under vaccination and vaccine non-responders prevents widespread immunity.

There are studies of HBV vaccination showing low adherence to multi-dose regimes which contributes to under vaccination (e.g. two studies report on this in the U.S: PMID: 35176102, PMID: 34097776). There is also the issue of availability of HBV vaccination in some countries. Furthermore, there is the issue of these vaccines not providing benefit to an already infected individual.

2. Reference 8 is a study regarding HCV-HBV co-infection and may not be the best reference for vaccine non-responders and consideration should instead be given studies such as the following example (PMID: 33238923).

Reference 9, instead suggests in some populations the vaccine non-responder rates could be as high as 16% (the referenced study of healthcare workers in Kenya) this should be reflected by the preceding statement if this is the evidence used.

3. Reference 20 is about exhausted CD8+ T cells but reference 21 is not an example of exhausted CD8+ T cell development in HBV. The point of this study and others (PMID: 38897196) is that the phenotype of HBV-antigen specific CD8+ T cells is distinct from the canonical 'exhaustion phenotype' previously described for other chronic viral infections and instead is described as a dysfunctional state for HBV-specific CD8+ T cells.

4. This manuscript would benefit from a more comprehensive presentation of the current knowledge regarding the ST2-IL-33 axis in HCC. IL-33 is known to expand Treg cells in the liver and previous studies have identified the involvement of IL-33 in liver diseases. The following references should be considered for inclusion in the discussion of the results to appropriately contextualise this study within the field (PMID: 35749514 & PMID: 29729112).

Limitations of the model used

5. The use of a HBV model in which the viral genome is integrated into hepatocytes via tail vein hydrodynamic injection is an inflammatory delivery mechanism which does not recapitulate the manifestation of HBV infection in humans as this is a non-cytopathic virus characterized by hiding from the immune system and minimal inflammation. Furthermore, this is not strictly an infection as the full virus is not replicating within cells. For these reasons the potential impact of this model on the interpretation of the results should be discussed in a limitations section.

6. DEN, once injected, and within the liver, is activated by cytochrome P450 enzymes and starts mediating DNA damage. It is acknowledged that no one model of HCC can accurately recapitulate the human manifestation of HCC, but this should be also be acknowledged in the limitations section.

7. Can the authors explain why the incidence of HCC in male B6 mice administered DEN at 4-5 weeks old seems low. It is acknowledged that this is very different kinetics to the standard use of this model (when mice are 2 weeks old). However, a single i.p. injection at 2 weeks is sufficient to result in 90% of mice developing HCC by 9 months. In this study the dose is escalated and for 8 consecutive weeks. It is concerning that there is minimal development of HCC even at 12 months. Some references showing similar protocols, or a more detailed explanation would be beneficial for the reader to understand the rationale.

Analytical approach: Statistical analysis

8. It is not clear whether each set of experiments have been repeated according to what is stated in the figure legends or reporting summary. If any of these data are from a single experiment it would be necessary to repeat them to ensure reproducibility of the results.

9. It is important that in the figure legends it is described whether each symbol represents a biological or technical replicate to describe the data shown appropriately. Authors should add this to their figure legends for clarity.

10. It is acknowledged that the authors have conducted appropriate statistical tests and the n number has been listed for each experimental group for each figure. P- values are consistently listed in the figures but there is inconsistency with whether non-significant P-values are displayed. Please decide whether to display only significant comparisons or both significant and non-significant and keep this consistent throughout the figures.

Data and methodology:

Flow cytometry

11. From the gating strategies shown, it appears that no exclusion of dead cells has been conducted on flow cytometry data. As dead cells are sticky it could account for why there are so many CD8 CD4 double-positive T cells in the plots in supplementary figure 2. It is recommended in the absence of the ideal scenario of viability dye usage, it would be beneficial to at least apply size exclusion based on forward and side scatter parameters to improve the reliability of flow cytometry data. This exclusion of dead cells is particularly important with restimulation of cells.

12. Representative staining of IL-10 is not shown in figures 2 and 3. In supplementary figure 3, the histograms depict IL-10 staining and again no dead cell exclusion has been applied. Is there an unstimulated control or a control without Golgi apparatus inhibitors that has been used as a negative control for cytokine production? In the absence of this, is there a negative control for IL-10 expression (can you show the CD8+ T cells- although they may also express IL-10?) Was this assessed?

Further experimental evidence

13. It would strengthen the rationale for choosing to investigate IL-33 if it is the most differentially expressed interleukin at the protein level in addition to the mRNA level. If supernatants could be used for analysis each of the different cytokines identified in the transcriptional analysis should be assayed for by ELISA and if tissue blocks are available, they could also be quantified using IHC. This would allow assessment of the most differentially expressed cytokines at the protein level.

14. It is interesting that unlike IL33, IL1b, IL4, IL16, and IL7, the cytokine IL18 is only expressed in DEN treated mice without the HBV viral genome in hepatocytes. Could the authors discuss why this might be the case?

15. In figure 2 f-h, authors have used splenic Treg cells in culture to show the effect of IL-2 and IL-33, does this make sense given the focus of the study on liver Treg cells? Could the authors repeat these experiments with liver Treg to determine if they respond in the same way to these cytokines?

16. In figure 3, the use of Treg ST2 CKO mice elucidated that loss of ST2 in Treg cells compromised their

immunosuppressive function. It is acknowledged that there is an observed increase in CD8 and total CD3 T cells with deletion of ST2 expression by Treg cells, however, it would be ideal to show that this increase corresponds to CD44hi activated effector T cells that are making some type of effector cytokine. This would be required to show there is a reduction in the suppressive capacity of these Treg. Also the confocal imaging section (figure 3g) is difficult to clearly see. Some arrows or zoomed in images to clarify the observed CD8+ T cells would improve readability.

17. Which liver cells are responsible for making the IL-33 that is driving the progression of HCC? Is it the infected hepatocytes, antigen presenting cells, or endothelial cells?

18. Is there evidence of increased tissue damage in the absence of IL-33? As blocking this pathway was shown to block the function of Treg cells, is there a reduction in tissue repair or alternatively an increase in tissue damage?

Minor comments:

19. It appears that there are some typos in the methods section. Centrifugation speeds of 2000 x g for removal of hepatocytes and 3000 xg for pelleting immune cells are very high.

20. The y-axis label for CD8 appears to have a typo in supplementary figures 2 and 3, it says BV396 when the plot above states BUV395 for CD8. This should be corrected.

21. If possible, it would also be beneficial to show the number of HCC lesions in each of the livers.

22. Authors should also list the post-test corrections used for their statistical analysis in each case in the figure legends.

Reviewer #3

(Remarks to the Author)

In this manuscript, Jong Ho Park et al, report on the impact of HBV on chemical-induced liver cancer, through modulation of immune response.

This manuscript follows a recent report entitled "statin prevents cancer development in chronic inflammation by blocking IL-33 expression" published by the same team in the journal Nature Communication (doi.org/10.1038/s41467-024-48441-8). After a brief introduction, the authors first focused on the sensitivity to DEN treatment in mice expressing or not HBV. Park et al, identified that HBV enhanced the effect of DEN in liver carcinogenesis, through severe chronic inflammation driven by IL-33 expression. Then, using different KO mouse models for IL-33 and ST2 genes, these authors suggested that IL-33 supports the activation of CD4+, ST2+ Treg cells and, subsequently, the expression of TGF- β and IL-10, which enhance cancer development.

Finally, Park et al, focused on the protective effect of Pitavastatin (lipid-lowering drug), which reduces the increased sensitivity to liver carcinogenesis in HBV-expressing DEN-treated mice. The translational aspect was also investigated by the quantification of IL-33 in serum samples from HBV-infected patients.

Although the observations described by Park et al, may be of interest regarding to the role of HBV in chemical-induced liver carcinogenesis, this study needs to be substantially amended.

Major modifications:

1) These authors claimed that the HBV genome, transferred by a transfection method using hydrodynamic injection (HDI), into mice at 1 month of age, remained persistent in this model. This persistence was illustrated (Fig.1) by the stable level of HBsAg expression in untreated mice. This data is largely insufficient to illustrate the viral genome persistence. At the very least, viral load, Viral RNA expression and Hbc staining in the liver should be illustrated in both untreated and DEN-treated mice.

2) In line with the last point, HDI transfection maintains the plasmid genome mainly as episomal forms within the nucleus of liver cells. In an inflammatory context, such as those induced by DEN treatment, the cytotoxicity is associated with a compensatory liver regeneration (PCNA expression). In this context HBV DNA plasmid will be removed from hepatocytes. It is therefore critical to demonstrate the persistence of HBV DNA in liver cells during and after DEN injections.

3) In the same line, Park et al, mentioned HBV infection throughout their description of the HBV mouse model. This model is clearly not a model of infection. Moreover, viral spreading was not possible. This concept should be modified in the amended version.

4) These experiments were conducted between four and twelve months after multiple injections of DEN. The liver tumor area at 12 months was never shown. In addition, the tumor area in wild-type mice 8 months post-DEN treatment varies considerably in figure 1 (1C) compared with the data shown in figure 2 (2B) and figure 3 (3B). Taking these variations into account, are the significant differences described in figure 1C still present?

5) Immunofluorescence illustrations should be optimized (Fig.1g; 3g) to better appreciate staining by, at least, a counterstaining to see the membrane of hepatocytes and CD45 cells.

6) Park et al reported increased HMGB1 staining in the liver of HBV+DEN mice. However, this staining was predominantly nuclear and not cytoplasmic, as usually described for this DAMP.

Minor modifications:

1) Tumor surface was expressed as a percentage of liver surface area. However, it is the volume of the tumor that should be taken into account, not just the surface area.

2) As previously reported, IL-33 can significantly reduce HBV DNA and HBsAg expression (Gao et al, 2020; Shen et al, 2017). This antiviral effect should be investigated in this study.

Version 1:

Reviewer comments:

Reviewer #1

(Remarks to the Author)

The authors adequately addressed the comments raised by the reviewers, and the contents have significantly improved.

Reviewer #2

(Remarks to the Author)

Post review, it is evident that Huang et al., have extensively addressed the points made by each of the reviewers. Overall, this study improves our understanding of HBV-associated HCC and identifies a ST2-Treg – IL-33 axis which exacerbates disease. The results suggest that inhibition of IL-33 signalling could provide therapeutic benefit in HBV-associated HCC.

Responses to comments:

Clarity and context:

The modifications to the manuscript have improved the contextualisation of these results within the existing literature and with reference to human disease. Furthermore, these changes have clarified the main points made by this study and now the conclusions drawn are more comprehensively supported by the data presented.

Limitations of the model used:

The addition of a section to focus on limitations of this study has improved the discussion and transparency of the work presented in this manuscript.

Addition of experiments:

The human scRNA-seq dataset showing an increase in ST2+ Treg in HBV-associated chronic hepatitis strengthens the data presented in the mouse model as both suggest the importance of this specific Treg population.

The additional quantification of liver IL-1 β , IL-4, and IL-16 has improved the rationale for focusing on IL-33 as it is clear now that it is altered in this model with respect to controls.

The T cell culture using liver Treg instead of splenic has improved the relevance of the results in figure 2. This is in agreement with existing literature suggesting that tissue Tregs will exhibit greater expression of the functional markers such as ST2 in comparison with their splenic counterparts.

The additional experimental evidence provided clarifies the points made and solidifies the rationale of this study.

Flow cytometry:

Following the response to reviewer 2, point 11 (with reference to flow cytometry). The authors have satisfactorily addressed the concerns.

It would be important to note for future experiments that although Percoll has been used to isolate cells and it is understood that dead cells will be removed with this processing, it is still always better to add a viability stain. This is because after Percoll washing is required prior to staining and small numbers of cells can die during the normal staining process. It was also stated in the methods that a cell-activator cocktail is used for restimulation of the cells. With restimulation there will also always be more cell death.

Statistical Analysis:

The addition of further details concerning the statistical tests used, exact group sizes and number of independent experiments has clarified the robust nature of the study and improved transparency for the reader.

Reviewer #3

(Remarks to the Author)

In the amended version of Huang et al manuscript's, they report on the impact of HBV on chemical-induced liver carcinogenesis (DEN), through secretion of HMGB1. This DAMP led to the activation of IL-33, supporting the stimulation of ST2+ Treg cells and subsequently, the expression of TGF- and IL-10, which enhanced liver cancer development. Huang et al also focused on the protective effect of Pitavastatin (targeting the IL-33/Treg axis), which reduced the sensitivity to liver carcinogenesis in HBV-expressing DEN-treated mice and in patients. Translational aspect was also investigated by

the quantification of ST2+ Tregs (scRNA seq data) in liver and IL-33 in serum samples from HBV-infected patients.

Despite the appreciated considerable efforts made by Huang et al, to answer the several questions raised by the 3 reviewers and thus improve this amended version, some points remained unresolved.

1) These authors claimed that HBV genome, transferred by a transfection method using hydrodynamic injection (HDI), in male mice at 1 month of age, remained persistent in the model of liver carcinogenesis (2-months of DEN injection). This viral persistence, previously illustrated by the stable level of HBsAg expression in untreated mice, is now also illustrated by HBc staining and quantification of HBV viral load (at 4 months post-transfection) in DEN-treated mice. However, these data have always been insufficient to validate the role of HBV viral persistence in the process of carcinogenesis and immune recruitment.

Indeed:

- The level of HBsAg was not illustrated in the blood of DEN-treated mice.
- The amount of HBV DNA, quantified in the serum samples of DEN-treated mice, was reported as a relative value (vs. positive control of the kit used) and not as viral load. Quantification in IU/ml gives a more accurate assessment of the level of viral expression (as illustrated for patients in Supplementary Table 1).
- Similarly, the number of HBc positive cells in the liver could be quantified for comparison to hepatic PCNA, IL-33 and HMGB1 staining.
- Finally, both additional virological data in this amended manuscript were performed only at 4-months post-transfection (without liver tumor). The same analyses might be completed at 8 months.

2) As previously suggested, transfection using hydrodynamic injection maintains the plasmid genome mainly in episomal form into the nucleus of liver cells. Huang et al showed that 20 to 80% of hepatocytes were positive for PCNA and/or IL_33 and/or HMGB1 staining at 4-months post-HDI. In this context, episomal HBV DNA could probably be rapidly eliminated from hepatocytes and, considering the absence of HBV spreading, the link between HBV and immune regulation is uncertain throughout the carcinogenesis process.

3) Park et al reported an increase of HMGB1 staining in the liver of HBV+DEN treated mice. This data was reinforced in this amended manuscript by the quantification of HMGB1 in liver and serum samples of treated mice. However, the serum quantification of this DAMP did not seem to be significantly different in presence or not of HBV in DEN-treated mice.

4) Finally, could the contribution of HBV to the “modulating the immune response to environmental carcinogens” be related only to the early stages of HCC?

5) Despite the better description of the model used by these authors, “HBV infection” is still mentioned in the amended version of this manuscript (such as line 101, 102, 113, p6....) and should be replaced by HBV expression or HBV infection-like, e.g.

Version 2:

Reviewer comments:

Reviewer #3

(Remarks to the Author)

The authors have considered the comments made during the last evaluation, and the manuscript content has been significantly refined.

Response to Reviewers' Comments:

We thank the reviewers for their insightful comments, which have improved our manuscript. Our replies to the comments are provided in blue below. Manuscript text and figures have also been revised (highlighted in blue) according to the reviewers' comments.

REVIEWER COMMENTS

Reviewer #1:

In this study, the authors demonstrated that HBV infection alone does not directly cause liver inflammation or cancer. Instead, they found that HBV alters the chronic inflammation induced by chemical carcinogens, such as DEN, to promote liver carcinogenesis. The combined administration of HBV and DEN activated the IL-33/regulatory T cell axis, which is essential for liver cancer development. Pitavastatin, used as an IL-33 inhibitor, successfully suppressed liver cancer induced by the HBV and DEN combination. The findings highlight the significant role of environmental carcinogens in increasing the risk of HBV-related hepatocellular carcinoma (HCC).

The authors propose that the combination of HBV infection and chemical carcinogens is crucial for driving chronic inflammation. However, it is well established that viral infections typically activate type I interferon (IFN) signaling, which contributes to inflammatory microenvironments. Therefore, it is recommended that the authors assess type I IFN signaling in their model. Additionally, many of the findings are derived from a mouse model, raising concerns about the clinical relevance to human HBV-associated HCC, which needs further investigation.

We thank the reviewer for their remarks and the critical points raised.

Major issues:

1. The authors identified cytokines such as IL-33, IL-1 β , IL-4, IL-16, and IL-7 as differentially expressed in AAV-infected, HBV-expressing hepatocytes. However, these cytokines are commonly expressed by various immune cells, not hepatocytes. Given the possible influence of hydrodynamic injection on cells around the liver sinusoid, including hepatic stellate cells, it is critical to confirm the specific cell types expressing these cytokines. IL-33, for instance, is known to be produced by hepatic stellate cells to suppress anti-tumor immunity (PMID: 35749514). The authors should use single-cell analysis or immunohistochemistry with cell-type-specific markers to verify the sources of these cytokines. Additionally, they should clarify why IL-18 expression was repressed (Figure 1j), considering that IL-1 β and IL-18 are activated via inflammasomes.

We thank the reviewer for raising this important point. In our studies, we have found IL-33 upregulation in HBV+DEN-treated liver compared with Sham+DEN-treated liver (Figure 1j). Considering the role of DAMPs and TLR signaling in IL-33 induction, hepatocytes, as well as other resident liver cells, including hepatic stellate cells, can be a source of IL-33^{1,2,3}. Nonetheless, as suggested, we examined which liver cell types upregulated cytokines in the carcinogen-treated liver. We found that IL-33 was expressed in liver epithelial cells and hepatic stellate cells (Figure R1a, b). In addition, we found IL-1 \$\beta\$ was expressed in liver epithelial cells and hepatic stellate cells, while IL-4 was expressed in hepatic stellate cells (Figure R1c-f). Notably, we did not find upregulation of IL-1 \$\beta\$ or IL-4 in HBV+DEN- compared with Sham+DEN-treated livers at the protein level (please see Figure R8 below). Thus, IL-33 is the main interleukin elevated in the liver of

HBV+DEN-treated mice. We have added these data to our revised manuscript (Supplementary Fig. 2b).

Figure R1. Cytokine expression in carcinogen-treated mouse liver. **a** Representative image of IL-33 and E-cadherin stained liver treated with carcinogenesis protocol at 4 months post-infection. The arrow points to IL-33 expressing hepatocyte. **b** Representative image of IL-33 and α -SMA stained liver treated with carcinogenesis protocol at 4 months post-infection. The arrow points to IL-33 expressing hepatic stellate cells. **c** Representative image of IL-1 β and E-cadherin stained liver treated with carcinogenesis protocol at 4 months post-infection. The arrow points to IL-1 β expressing hepatocyte. **d** Representative image of IL-1 β and α -SMA stained liver treated with carcinogenesis protocol at 4 months post-infection. The arrow points to IL-1 β expressing hepatic stellate cells. **e** Representative image of IL-4 and E-cadherin stained liver treated with carcinogenesis protocol at 4 months post-infection. **f** Representative image of IL-4 and α -SMA stained liver treated with carcinogenesis protocol at 4 months post-infection. The arrow points to IL-4 expressing hepatic stellate cells. Scale bars: 100 μ m.

2. While the authors show that hepatocyte-derived IL-33 stimulates ST2-positive Treg cells, it is important to determine whether this occurs in human HBV infections as well. The authors should assess the prevalence of ST2-positive Treg cells in human HBV-infected liver tissues, as the mouse model may not fully capture the human immune environment.

We thank the reviewer for raising this critical point. To examine the prevalence of ST2⁺ Tregs in human liver, we analyzed single-cell RNA sequencing (scRNA-seq) data (GSE234241)⁴ and detected ST2⁺ Tregs in HBV-associated chronic hepatitis, which was increased compared with healthy control liver (Figure R2a). In addition, we analyzed single-cell RNA sequencing data (GSE182159)⁵ of immune cells from HBV-associated immune active (IA), chronic resolved (CR), and HBV-free healthy controls (HC). The IA phase represents active liver inflammation, fibrosis, and HBV DNA positivity. ST2⁺ Tregs were increased in IA compared with CR and HC (Figure R2b, c). These findings support the role of ST2⁺ Tregs in HBV-associated hepatitis in humans. We have added these data to our revised manuscript (Supplementary Fig. 5).

Figure R2. ST2⁺ Tregs are increased in HBV-associated chronic hepatitis in humans. **a** scRNA-seq UMAP plot of ST2⁺ Tregs (cyan dots) among T cells in human liver tissues from HBV-associated chronic hepatitis (n = 13) and healthy controls with no viral markers (n = 3). There are 175 ST2⁺ Tregs out of 16021 T cells in HBV hepatitis (1.1%) and 18 ST2⁺ Treg cells out of 2178 T cells in healthy control (0.8%). **b** scRNA-seq UMAP plots of Tregs (red dots) among CD4⁺ T cells in human liver tissues from HBV-associated hepatitis (immune active or IA, n = 5), chronic resolved hepatitis (CR, HBV DNA level is below the detection limit, n = 3), and HBV-free healthy controls (HC, n = 6). **c** *IL1RL1* expression highlights ST2⁺ CD4⁺ T cells (blue dots). The arrow points to the Treg cluster.

3. The authors propose that HBV infection, combined with carcinogen exposure, leads to hepatocyte damage and DAMP release, triggering IL-33 expression. This hypothesis is intriguing but requires further investigation. The authors should specify which cell types express IL-33 upon DAMP stimulation. Moreover, given the variety of carcinogens in natural environments, what is the common molecular mechanism driving the second stimulation? Is it linked to DNA damage, necrotic cell death, or another process? Clarifying the immunological consequences of this chronic inflammation is crucial for understanding these findings.

We appreciate this point. Previous studies have shown that DEN induces immunogenic cell death in hepatocytes^{6,7}. In addition, HMGB1 is a key DAMP molecule associated with immunogenic cell

death^{8,9}. Thus, DEN induces hepatocyte damage, which can lead to HMGB1 release. HMGB1 from damaged hepatocytes can in turn trigger the TLR4 signaling pathway activation in the neighboring cells to drive IL-33 expression¹⁰. To address the reviewer's comment, we examined the impact of DEN on a mouse hepatocyte cell line (AML12)¹¹. DEN treatment led to an increase in the cell death checkpoint molecule p53 and the apoptosis marker cleaved PARP1 in AML12 cells (Figure R3).

Figure R3. DEN induces immunogenic cell death in hepatocytes. Dose-dependent increase in cleaved-PARP1 and p53 protein expression in AML12 cells after DEN treatment. GAPDH is used as the house keeping protein control.

Furthermore, we detected increased DNA damage marked by γ H2AX in DEN-treated liver (Figure R4). Importantly, HBV+DEN treatment led to a significantly higher number of γ H2AX⁺ cells in the liver compared with Sham+DEN, supporting our finding that increased DNA damage and DAMP release drive IL-33 expression in HBV-infected livers exposed to a carcinogen. We have added these findings to the revised manuscript (Supplementary Fig. 3b, c).

Figure R4. HBV plus carcinogen exposure promotes DNA damage in the liver. **a** Representative images of γ H2AX stained HBV+DEN-, Sham+DEN-, and Sham+PBS-treated liver at 4 months post-infection. **b** Quantification of γ H2AX⁺ cells per 100 cells in the HBV+DEN (n = 7 mice) compared with Sham+DEN- (n = 8 mice) and Sham+PBS-treated liver (n = 6 mice) at 4 months post-infection. γ H2AX⁺ cells per 100 cells were counted in ten randomly selected HPF images per liver. Each dot represents an HPF image. Graph shows mean + SD, one-way ANOVA with Tukey's multiple comparison test, Scale bar: 100 μ m.

4. The serum IL-33 levels in hepatitis patients (n=12, Figure 4f) are significantly elevated regardless of HBV infection. However, the sample size is too small to draw definitive conclusions about differences between HBV-positive and HBV-negative HCC patients. The authors should increase the sample size to strengthen their findings. Additionally, they should investigate the mechanism behind elevated IL-33 levels in non-HBV patients, as this could provide insight into the mechanism of IL-33 induction in liver diseases. This needs to be mentioned in the text as a caveat to data interpretation.

As suggested by the reviewer, we increased the sample size for human serum IL-33 level determination in HBV hepatitis group (n = 40), non-HBV hepatitis group (n = 40), and healthy controls (n = 24). Based on the additional samples analyzed, we found that IL-33 levels were upregulated in HBV hepatitis compared with non-HBV hepatitis and healthy controls (Figure R5). Although not statistically significant, IL-33 levels trended higher in non-HBV hepatitis compared with healthy controls. We speculate that cellular damage and DAMPs release in non-HBV hepatitis may stimulate IL-33 expression. We have added these data to our revised manuscript (Fig. 4f).

Figure R5. Serum IL-33 levels are increased in HBV-positive hepatitis patients. Serum IL-33 levels in HBV-positive hepatitis patients (n = 40), HBV-negative hepatitis patients (n = 40), and healthy controls (n = 24). Graphs show mean + SD, one-way ANOVA with Tukey's multiple comparison test.

5. The use of pitavastatin as an inhibitor of IL-33-mediated pathways raises concerns. Pitavastatin is primarily an inhibitor of HMG-CoA reductase, which reduces cholesterol synthesis. The authors need to clarify how inhibiting HMG-CoA reductase impacts IL-33 expression. The reduced incidence of HCC might be related to lowered cholesterol levels, rather than direct inhibition of IL-33. Further mechanistic studies are required to bridge the gap between cholesterol metabolism and the IL-33 signaling pathway.

We appreciate this important comment. In our previous publication¹⁰, we have demonstrated the cholesterol-independent mechanism by which pitavastatin suppresses IL-33 expression in epithelial cells. We have found that pitavastatin blocks TBK1 membrane binding, which is the critical step in TBK1 phosphorylation and activation. In turn, diminished phospho-TBK1 reduces IRF3 activity, which is the major transcription factor driving IL-33 expression in chronic inflammation. We have clarified the pitavastatin mechanism of action in our revised manuscript (Discussion, page 12, first paragraph).

Reviewer #2:

In this study, the authors detail an investigation of the link between chronic hepatitis B virus (HBV) genome delivery to hepatocytes and the development of hepatocellular carcinoma (HCC). The HBV genome was delivered to hepatocytes via hydrodynamic injection of pAAV-HBV1.2, which alone was not sufficient for HCC development. It was determined that pAAV-HBV1.2 delivery combined with carcinogen administration (diethyl nitrosamine: DEN) increased HCC incidence and progression and as a result reduced survival compared to empty vector AAV mice treated with DEN. In mice deficient in ST2, IL-33, or IRF3, HBV+DEN treatment resulted in a significant reduction in HCC development. The progression of HCC was dependent on expansion of ST2+ Tregs via IL-33 production which was enhanced in HBV infected, DEN-treated mice. Inhibiting signalling through ST2 via Pitavastatin treatment resulted in reduced HCC development and improved survival. The results of this manuscript address an important knowledge gap regarding the development of HCC in patients with chronic HBV.

There are some major points for improvement of this manuscript including further contextualisation within the current literature, appropriate discussion of the limitations of the study, increased clarity regarding repetition of experiments, and improvement of flow cytometry data. These major concerns should be addressed to improve the scientific rigor and quality of this study.

Key results:

- Authors show that chronic HBV infection changes the response to chemical carcinogens, as in their model, HBV genome expression in hepatocytes exacerbated the development of HCC.

- The increased HCC development in AAV-HBV-infected mice was shown to be dependent on ST2 expression by Treg cells.

- Inhibition of ST2-IL-33 axis results in reduced liver tumour burden in mice with the HBV genome in hepatocytes treated with a carcinogen.

Significance & validity:

Authors have identified a therapeutic strategy for patients with HBV and HCC. Studies focused on HCC in isolation have previously identified an important role in IL-33 signalling in the progression of disease potentially limiting the significance of this part of the story. However, this study expands on the current knowledge by addressing the link between HBV-genome expressing hepatocytes and HCC. The authors demonstrate reduced tumour burden upon inhibition of IL-33-ST2 signalling axis in their model.

We thank the reviewer for their remarks and the critical points raised.

Major comments**Clarity and context**

The Introduction requires careful rephrasing

1. Lines 49-53 could be re-phrased to more clearly state the current challenges in ensuring HBV immunity, it would be more precise to state a combination of factors contribute to the global threat of HBV:

Although prophylactic vaccination is available, a combination of under vaccination and vaccine non-responders prevents widespread immunity.

There are studies of HBV vaccination showing low adherence to multi-dose regimes which contributes to under vaccination (e.g. two studies report on this in the U.S: PMID: 35176102, PMID: 34097776). There is also the issue of availability of HBV vaccination in some countries.

Furthermore, there is the issue of these vaccines not providing benefit to an already infected individual.

We appreciate this important comment. We have revised the manuscript to elaborate on these points (Introduction, page 3, first paragraph).

2. Reference 8 is a study regarding HCV-HBV co-infection and may not be the best reference for vaccine non-responders and consideration should instead be given studies such as the following example (PMID: 33238923).

Reference 9, instead suggests in some populations the vaccine non-responder rates could be as high as 16% (the referenced study of healthcare workers in Kenya) this should be reflected by the preceding statement if this is the evidence used.

We appreciate this important comment. We have revised the manuscript to elaborate on these points (Introduction, page 3, first paragraph).

3. Reference 20 is about exhausted CD8+ T cells but reference 21 is not an example of exhausted CD8+ T cell development in HBV. The point of this study and others (PMID: 38897196) is that the phenotype of HBV-antigen specific CD8+ T cells is distinct from the canonical 'exhaustion phenotype' previously described for other chronic viral infections and instead is described as a dysfunctional state for HBV-specific CD8+ T cells.

We thank the reviewer for raising this important point. We have revised the manuscript accordingly (Introduction, page 3, second paragraph).

4. This manuscript would benefit from a more comprehensive presentation of the current knowledge regarding the ST2-IL-33 axis in HCC. IL-33 is known to expand Treg cells in the liver and previous studies have identified the involvement of IL-33 in liver diseases. The following references should be considered for inclusion in the discussion of the results to appropriately contextualise this study within the field (PMID: 35749514 & PMID: 29729112).

We appreciate this point. We have added the recommended references to our revised manuscript (Discussion section, page 12, second paragraph).

Limitations of the model used

5. The use of a HBV model in which the viral genome is integrated into hepatocytes via tail vein hydrodynamic injection is an inflammatory delivery mechanism which does not recapitulate the manifestation of HBV infection in humans as this is a non-cytopathic virus characterized by hiding from the immune system and minimal inflammation. Furthermore, this is not strictly an infection as the full virus is not replicating within cells. For these reasons the potential impact of this model on the interpretation of the results should be discussed in a limitations section.

We thank the reviewer for raising this point. We have added this limitation to our revised manuscript (Discussion section, page 13, second paragraph).

6. DEN, once injected, and within the liver, is activated by cytochrome P450 enzymes and starts mediating DNA damage. It is acknowledged that no one model of HCC can accurately recapitulate the human manifestation of HCC, but this should be also be acknowledged in the limitations section.

As suggested, we have added the limitation of experiments to our revised manuscript (Discussion section, page 13, second paragraph).

7. Can the authors explain why the incidence of HCC in male B6 mice administered DEN at 4-5 weeks old seems low. It is acknowledged that this is very different kinetics to the standard use of this model (when mice are 2 weeks old). However, a single i.p. injection at 2 weeks is sufficient to result in 90% of mice developing HCC by 9 months. In this study the dose is escalated and for 8 consecutive weeks. It is concerning that there is minimal development of HCC even at 12 months. Some references showing similar protocols, or a more detailed explanation would be beneficial for the reader to understand the rationale.

We thank the reviewer for raising this point. Although DEN administration at 2 weeks of age is sufficient to result in 90% of mice developing HCC by 9 months, we administered DEN to animals after the HBV carrier state was established at 4 weeks of age. A single dose of DEN injection in adult mice is not sufficient for HCC development. Therefore, we followed consecutive injection of DEN as previously described¹².

Analytical approach: Statistical analysis

8. It is not clear whether each set of experiments have been repeated according to what is stated in the figure legends or reporting summary. If any of these data are from a single experiment it would be necessary to repeat them to ensure reproducibility of the results.

We thank the reviewer for pointing this out. We have added this information to figure legends.

9. It is important that in the figure legends it is described whether each symbol represents a biological or technical replicate to describe the data shown appropriately. Authors should add this to their figure legends for clarity.

We have added this information to figure legends.

10. It is acknowledged that the authors have conducted appropriate statistical tests and the n number has been listed for each experimental group for each figure. P- values are consistently listed in the figures but there is inconsistency with whether non-significant P-values are displayed. Please decide whether to display only significant comparisons or both significant and non-significant and keep this consistent throughout the figures.

We thank the reviewer for pointing this out. We have revised the figures to consistently remove non-significant *p*-values, except in a few graphs where the demonstration of non-significant *p*-values was deemed essential to convey the intended message.

Data and methodology:

Flow cytometry

11. From the gating strategies shown, it appears that no exclusion of dead cells has been conducted on flow cytometry data. As dead cells are sticky it could account for why there are so many CD8 CD4 double-positive T cells in the plots in supplementary figure 2. It is recommended in the absence of the ideal scenario of viability dye usage, it would be beneficial to at least apply size exclusion based on forward and side scatter parameters to improve the reliability of flow cytometry data. This exclusion of dead cells is particularly important with restimulation of cells.

We appreciate this point. We used the Percoll protocol to isolate the lymphocytes from the liver for flow analysis. Thus, dead cells were already separated based on Percoll density¹³. To address the reviewer's comment, we revised the FSC/SSC gating to exclude the dead cells. The results were consistent with our original findings (Figure R6). We have revised the manuscript accordingly (Fig. 2e and Supplementary Fig. 4b).

Figure R6. ST2⁺ Tregs are reduced in HBV+DEN-treated I133^{KO} liver. **a** Flow gating strategy for determining ST2⁺ immune cell percentages in the liver. Numbers on the flow plots represent % cells in the gate. **b** Percent ST2⁺ immune cell types in I133^{KO} (n = 6) and WT liver (n = 4) treated with HBV+DEN at 4 months post-infection. Each dot represents a mouse. Graphs show mean + SD, two-sided unpaired t-test.

12. Representative staining of IL-10 is not shown in figures 2 and 3. In supplementary figure 3, the histograms depict IL-10 staining and again no dead cell exclusion has been applied. Is there an unstimulated control or a control without Golgi apparatus inhibitors that has been used as a negative control for cytokine production? In the absence of this, is there a negative control for IL-10 expression (can you show the CD8⁺ T cells- although they may also express IL-10?) Was this assessed?

We appreciate these points. As stated above, the Percoll protocol used for isolating lymphocytes from the liver removes most dead cells. Nonetheless, we revised the FSC/SSC gating to further exclude the dead cells, which did not affect the outcomes (Figure R7). In addition, we added the IL-10 expression data on CD8⁺ T cells and unstimulated Tregs as controls (Figure R7). We have revised our manuscript accordingly (Fig. 3f and Supplementary Fig. 6b-d).

Figure R7. IL-10-expressing Tregs are reduced in TregST2^{CKO} liver. **a** Flow gating strategy to identify IL-10-expressing Treg population in the liver. **b** IL-10⁺ Treg frequency in TregST2^{CKO} (n = 7) and WT liver (n = 7) treated with HBV+DEN at 4 months post-infection. **c** Representative histograms of Tregs' IL-10 expression in HBV+DEN-treated TregST2^{CKO} and WT liver at 4 months post-infection. **d** IL-10⁺ CD8⁺ T frequency in TregST2^{CKO} (n = 7) and WT liver (n = 7) treated with HBV+DEN at 4 months post-infection. **e** Representative histograms of CD8⁺ T cells' IL-10 expression in HBV+DEN-treated TregST2^{CKO} and WT liver at 4 months post-infection. **f** Representative histograms of Tregs' IL-10 expression in samples not treated with cell activator cocktail (unstimulated control) from HBV+DEN-treated TregST2^{CKO} and WT liver

at 4 months post-infection. Each dot represents a mouse. Graphs show mean + SD, two-sided unpaired *t*-test.

Further experimental evidence

13. It would strengthen the rationale for choosing to investigate IL-33 if it is the most differentially expressed interleukin at the protein level in addition to the mRNA level. If supernatants could be used for analysis each of the different cytokines identified in the transcriptional analysis should be assayed for by ELISA, and if tissue blocks are available, they could also be quantified using IHC. This would allow assessment of the most differentially expressed cytokines at the protein level.

We appreciate these valuable points. As suggested by the reviewer, we examined the expression of other cytokines, including IL-1 β , IL-4, and IL-16, between HBV+DEN and Sham+DEN treatment groups (Figure R8). Unlike IL-33, these cytokines did not show significant differences at the protein level between the two groups. We have revised our manuscript accordingly (Supplementary Fig. 2c-e).

Figure R8. IL-1 β , IL-4, and IL-16 protein levels are not increased in HBV+DEN-treated WT liver. **a** IL-1 β protein levels in HBV+DEN- (n = 10 mice) versus Sham+DEN-treated liver (n = 9 mice) at 4 months post-infection. **b** IL-4 protein levels in HBV+DEN- (n = 10 mice) versus Sham+DEN-treated liver (n = 9 mice) at 4 months post-infection. **c** IL-16 protein levels in HBV+DEN- (n = 10 mice) versus Sham+DEN-treated liver (n = 9 mice) at 4 months post-infection. Graphs show mean + SD, two-sided unpaired *t*-test.

14. It is interesting that unlike IL33, IL1b, IL4, IL16, and IL7, the cytokine IL18 is only expressed in DEN treated mice without the HBV viral genome in hepatocytes. Could the authors discuss why this might be the case?

We thank the reviewer for raising this point. Unlike IL-33, IL-18 is constitutively expressed¹⁴. We hypothesize IL-18 regulation in the liver is distinct from IL-33. HBV+DEN-induced damaged cells preferentially express IL-33 but not IL-18 through activation of the TBK1/IRF3 signaling pathway¹⁰. However, determining the mechanism that suppresses IL-18 expression in HBV+DEN-treated liver requires future investigations. We have revised our manuscript to acknowledge the differential regulation of IL-18 in our model and the need for future investigations to determine its role in tumor development in HBV-associated HCC (Discussion section, page 13, first paragraph).

15. In figure 2 f-h, authors have used splenic Treg cells in culture to show the effect of IL-2 and IL-33, does this make sense given the focus of the study on liver Treg cells? Could the authors

repeat these experiments with liver Treg to determine if they respond in the same way to these cytokines?

We appreciate this critical point. As suggested, we examined the impact of IL-33 on liver Tregs. Similar to splenic Tregs, IL-33 addition to IL-2 led to upregulation of TGFβ and IL-10 in ST2⁺ Tregs (Figure R9). We have added these data to the revised manuscript (Fig. 2f-2h).

Figure R9. IL-33 upregulates TGF-β1 and IL-10 expression in hepatic Tregs. **a** Treg frequency as % total WT hepatic CD4⁺ T cells after incubation with IL-2 plus IL-33, IL-2, versus no cytokine treatment (PBS) control (n = 6). **b** TGF-β1 mean fluorescence intensity (MFI) of IL-2 plus IL-33, IL-2, versus PBS-treated ST2⁺ hepatic Tregs (n = 6). **c** IL-10 MFI of IL-2 plus IL-33, IL-2, versus PBS-treated ST2⁺ hepatic Tregs (n = 6). **d** TGF-β1 MFI of IL-2 versus PBS-treated ST2⁻ hepatic Tregs (n = 6). **e** IL-10 MFI of IL-2 versus PBS-treated ST2⁻ hepatic Tregs (n = 6). Each dot represents a mouse. Graphs show mean + SD, a, b, c: one-way ANOVA with Tukey's multiple comparison test, d, e: two-sided unpaired t-test

16. In figure 3, the use of Treg ST2 CKO mice elucidated that loss of ST2 in Treg cells compromised their immunosuppressive function. It is acknowledged that there is an observed increase in CD8 and total CD3 T cells with deletion of ST2 expression by Treg cells, however, it would be ideal to show that this increase corresponds to CD44^{hi} activated effector T cells that are making some type of effector cytokine. This would be required to show there is a reduction in the suppressive capacity of these Treg. Also the confocal imaging section (figure 3g) is difficult to clearly see. Some arrows or zoomed in images to clarify the observed CD8⁺ T cells would improve readability.

We appreciate this point. As suggested, we stained liver samples for CD44⁺ activated T cells and found that CD44⁺CD8⁺ T cells were significantly increased in HBV+DEN-treated Treg-ST2^{CKO} compared with WT liver (Figure R10). We have added this data to our revised manuscript (Supplementary Fig. 6e, f).

Figure R10. Activated CD44⁺CD8⁺ T cells are increased in HBV+DEN-treated TregST2^{CKO} liver. a Representative images of CD44 and CD8 stained TregST2^{CKO} and WT liver treated with HBV+DEN at 4 months post-infection. **b** Quantification of CD44⁺CD8⁺ T cells in TregST2^{CKO} (n = 4) and WT liver (n = 3) treated with HBV+DEN at 4 months post-infection. Note that all CD8⁺ cells are CD3⁺ T cells. T cells were counted in four randomly selected HPF images per liver sample. Each dot represents an HPF image. Graphs show mean + SD, two-sided unpaired *t*-test, scale bar: 100 μ m.

17. Which liver cells are responsible for making the IL-33 that is driving the progression of HCC? Is it the infected hepatocytes, antigen presenting cells, or endothelial cells?

We thank the reviewer for raising this point. In our studies, we have found IL-33 upregulation in HBV+DEN-treated liver compared with Sham+DEN-treated liver (Figure 1j). Considering the role of DAMPs and TLR signaling in IL-33 induction, hepatocytes, as well as other resident liver cells, including hepatic stellate cells, can be a source of IL-33^{3, 15}. Nonetheless, as suggested, we examined which liver cell types upregulated cytokines in HBV+DEN-treated liver. We found that IL-33 was expressed in liver epithelial cells and hepatic stellate cells (Figure R1a, b above). We have added these data to our revised manuscript (Supplementary Fig. 2b).

18. Is there evidence of increased tissue damage in the absence of IL-33? As blocking this pathway was shown to block the function of Treg cells, is there a reduction in tissue repair or alternatively an increase in tissue damage?

We appreciate this point. We did not observe any macroscopic or histological sign of tissue damage in TregST2^{CKO} and Il33^{KO} compared with WT liver treated with HBV+DEN at 8 months post-infection.

Minor comments:

19. It appears that there are some typos in the methods section. Centrifugation speeds of 2000 x g for removal of hepatocytes and 3000 xg for pelleting immune cells are very high.

We have revised the Methods section accordingly.

20. The y-axis label for CD8 appears to have a typo in supplementary figures 2 and 3, it says BV396 when the plot above states BUV395 for CD8. This should be corrected.

We have corrected this mistake.

21. If possible, it would also be beneficial to show the number of HCC lesions in each of the livers.

As suggested, we counted tumor numbers with all the mouse experiments. Tumor counts per liver data are consistent with tumor area as % liver surface shown in the manuscript (Figure R11). We have added these data to the revised manuscript (Supplementary Fig. 1b, 4a, 6a, and 7b).

Figure R11. Liver tumor counts across liver carcinogenesis studies. **a** Liver tumor counts in WT mice that underwent liver carcinogenesis protocol at 8 months post-infection. $n = 7$ mice in HBV+DEN group, $n = 8$ mice in Sham+DEN group, $n = 7$ mice in HBV+PBS group, and $n = 7$ mice in Sham+PBS group. **b** Liver tumor counts in Il33^{KO} ($n = 7$), ST2^{KO} ($n = 5$), Irf3^{KO} ($n = 6$), and WT mice ($n = 7$) that received HBV+DEN at 8 months post-infection. **c** Liver tumor counts in TregST2^{CKO} ($n = 12$) and WT mice ($n = 6$) that received HBV+DEN at 8 months post-infection. **d** Liver tumor counts in pitavastatin- ($n = 10$) and PBS-treated ($n = 5$) WT mice that received HBV+DEN at 8 months post-infection. Each dot represents a mouse. Graphs show mean + SD, a, b: one-way ANOVA with Tukey's multiple comparison test, c, d: two-sided unpaired *t*-test.

22. Authors should also list the post-test corrections used for their statistical analysis in each case in the figure legends.

We have added this information to figure legends.

Reviewer #3:

In this manuscript, Jong Ho Park et al, report on the impact of HBV on chemical-induced liver cancer, through modulation of immune response. This manuscript follows a recent report entitled “statin prevents cancer development in chronic inflammation by blocking IL-33 expression” published by the same team in the journal Nature Communication (doi.org/10.1038/s41467-024-48441-8). After a brief introduction, the authors first focused on the sensitivity to DEN treatment in mice expressing or not HBV. Park et al, identified that HBV enhanced the effect of DEN in liver carcinogenesis, through severe chronic inflammation driven by IL-33 expression. Then, using different KO mouse models for IL-33 and ST2 genes, these authors suggested that IL-33 supports the activation of CD4+, ST2+ Treg cells and, subsequently, the expression of TGF- β and IL-10, which enhance cancer development. Finally, Park et al, focused on the protective effect of Pitavastatin (lipid-lowering drug), which reduces the increased sensitivity to liver carcinogenesis in HBV-expressing DEN-treated mice. The translational aspect was also investigated by the quantification of IL-33 in serum samples from HBV-infected patients. Although the observations described by Park et al, may be of interest regarding to the role of HBV in chemical-induced liver carcinogenesis, this study needs to be substantially amended.

We thank the reviewer for their constructive comments.

1. These authors claimed that the HBV genome, transferred by a transfection method using hydrodynamic injection (HDI), into mice at 1 month of age, remained persistent in this model. This persistence was illustrated (Fig.1) by the stable level of HBsAg expression in untreated mice. This data is largely insufficient to illustrate the viral genome persistence. At the very least, viral load, Viral RNA expression and HBc staining in the liver should be illustrated in both untreated and DEN-treated mice.

We thank the reviewer for raising these important comments. To address these, we first measured HBV DNA levels in the serum of the mice that received the HBV genome at 4 months post-hydrodynamic injection using qPCR (Figure R12a). Next, we detected Hepatitis B core antigen (HBcAg) in HBV+DEN and HBV+PBS-treated WT liver at 4 months post-hydrodynamic injection (Figure R12b). These findings have been added to the revised manuscript (Supplementary Fig. 1d and 1e).

Figure R12. HBV genome persists in mouse liver for several months post-hydrodynamic pAAV-HBV1.2 injection. **a** HBV DNA levels in the serum of mice treated with HBV+DEN ($n = 10$) and HBV+PBS ($n = 4$) at 4 months post-hydrodynamic pAAV-HBV1.2 injection. HBV DNA positive and negative control samples are provided in the detection kit. Each dot represents a mouse. Graph shows mean + SD. **b** Representative images of HBcAg stained liver tissues from HBV+DEN-, HBV+PBS-, Sham+DEN-, and Sham+PBS-treated mice at 4 months post-hydrodynamic pAAV-HBV1.2 injection. Scale bar: 100 μm .

2. In line with the last point, HDI transfection maintains the plasmid genome mainly as episomal forms within the nucleus of liver cells. In an inflammatory context, such as those induced by DEN treatment, the cytotoxicity is associated with a compensatory liver regeneration (PCNA expression). In this context HBV DNA plasmid will be removed from hepatocytes. It is therefore critical to demonstrate the persistence of HBV DNA in liver cells during and after DEN injections.

We appreciate this critical point and have generated data demonstrating the persistence of the HBV genome in the liver after DEN treatment (Figure R12 above).

3. In the same line, Park et al, mentioned HBV infection throughout their description of the HBV mouse model. This model is clearly not a model of infection. Moreover, viral spreading was not possible. This concept should be modified in the amended version.

As suggested, we have revised our manuscript to clarify that our model mimics HBV infection while stating its limitations (Introduction, page 4, second paragraph, Result section, page 6, first paragraph, and Discussion section, page 13, second paragraph).

4. These experiments were conducted between four and twelve months after multiple injections of DEN. The liver tumor area at 12 months was never shown. In addition, the tumor area in wild-type mice 8 months post-DEN treatment varies considerably in figure 1 (1C) compared with the data shown in figure 2 (2B) and figure 3 (3B). Taking these variations into account, are the significant differences described in figure 1C still present?

We appreciate this valuable point. The HCC model used in our studies is based on exposure to a chemical carcinogen, which is destined to generate tumor outcomes with considerable variation^{16, 17}. However, we view this to strengthen our findings, which point to an essential mechanism in HCC development, overcoming any baseline variation in each group. To further ensure that within-group variation did not critically affect findings across our experiments, we analyzed the data with a new comparison between all HBV+DEN-treated WT mice (From Fig. 1c, 2b, and 3b) and only DEN-treated WT mice (From Fig. 1c). HBV+DEN-treated mice had

significantly more liver tumor area and tumor numbers compared with only Sham+DEN-treated mice (Figure R13).

Figure R13. Comparison of the tumor area and count of HBV+DEN-treated WT mice across the studies compared with Sham+DEN-treated mice. a Tumor burden measured as % liver surface of WT mice that underwent liver carcinogenesis protocol at 8 months post-infection. **b** tumor count of WT mice that underwent liver carcinogenesis protocol at 8 months post-infection. n=20 mice in HBV+DEN group, n=8 mice in Sham+DEN group. Graphs show mean + SD, two-sided unpaired *t*-test.

5. Immunofluorescence illustrations should be optimized (Fig. 1g; 3g) to better appreciate staining by, at least, a counterstaining to see the membrane of hepatocytes and CD45 cells.

We appreciate this point. We have revised our manuscript to clearly demonstrate the staining pattern of immune cells in the liver. In addition, we provide co-staining here to distinguish hepatocytes from immune cells (Figure R14).

Figure R14. Immune cell staining is distinct from hepatocytes in the liver. **a** Representative images of CD45 and E-cadherin stained liver tissues treated with carcinogenesis protocol at 4 months post-infection. **b** Representative images of CD8, CD3, and E-cadherin stained TregST2^{CKO} and WT liver treated with HBV+DEN at 4 months post-infection. Arrows point to CD8⁺ T cells in the liver. Scale bars: 100 μ m.

6. Park et al reported increased HMGB1 staining in the liver of HBV+DEN mice. However, this staining was predominantly nuclear and not cytoplasmic, as usually described for this DAMP.

We thank the reviewer for raising this point. HMGB1 is known to be expressed in the nucleus^{18, 19}. However, we agree that HMGB1 release from the cells denotes its role as a DAMP. Hence, we evaluated HMGB1 levels in the liver lysates and the sera of the mice, which showed highly elevated levels in HBV+DEN-treated mice (Supplementary Fig. 3d and f). We have clarified this point in the revised manuscript (Results section, page 7, first paragraph).

Minor points:

1. Tumor surface was expressed as a percentage of liver surface area. However, it is the volume of the tumor that should be taken into account, not just the surface area.

We appreciate this point. Although tumor area as a percentage of the liver surface area is commonly used for liver cancer assessment in mice^{20, 21, 22}, we analyzed tumor counts in each liver as a secondary outcome measure to validate our findings (Figure R11 above).

2. As previously reported, IL-33 can significantly reduce HBV DNA and HBsAg expression (Gao et al, 2020; Shen et al, 2017). This antiviral effect should be investigated in this study.

We thank the reviewer for raising this point. In our study, we found that HBV alone does not cause liver cancer development. However, IL-33 induction in HBV+DEN-treated animals generated chronic inflammation and promoted Tregs to create a tumor-promoting immune environment in the liver. A previous study has also shown that IL-33 induces the proliferation of Treg in the liver²³. Thus, the anti-viral function of baseline IL-33 expression in the liver may not reflect its broader pro-tumorigenic function when induced upon carcinogen exposure. Nonetheless, we have discussed the impact of IL-33 on HBV expression in our revised manuscript (Discussion section, page 12, second paragraph).

References

1. Wang, N. *et al.* Toll-like receptor 3 mediates establishment of an antiviral state against hepatitis C virus in hepatoma cells. *J Virol* **83**, 9824-9834 (2009).
2. Jia, L. *et al.* Hepatocyte toll-like receptor 4 deficiency protects against alcohol-induced fatty liver disease. *Mol Metab* **14**, 121-129 (2018).
3. Stoss, C. *et al.* TLR3 promotes hepatocyte proliferation after partial hepatectomy by stimulating uPA expression and the release of tissue-bound HGF. *FASEB J* **34**, 10387-10397 (2020).
4. Genshaft, A.S. *et al.* Single-cell RNA sequencing of liver fine-needle aspirates captures immune diversity in the blood and liver in chronic hepatitis B patients. *Hepatology* **78**, 1525-1541 (2023).
5. Zhang, C. *et al.* Single-cell RNA sequencing reveals intrahepatic and peripheral immune characteristics related to disease phases in HBV-infected patients. *Gut* **72**, 153-167 (2023).
6. You, Y. *et al.* Phyllanthin prevents diethylnitrosamine (DEN) induced liver carcinogenesis in rats and induces apoptotic cell death in HepG2 cells. *Biomed Pharmacother* **137**, 111335 (2021).
7. Naylor, G. *et al.* Immunogenic Death of Hepatocellular Carcinoma Cells in Mice Expressing Caspase-Resistant ROCK1 Is Not Replicated by ROCK Inhibitors. *Cancers (Basel)* **14** (2022).
8. Zhou, J. *et al.* Immunogenic cell death in cancer therapy: Present and emerging inducers. *J Cell Mol Med* **23**, 4854-4865 (2019).
9. Fucikova, J. *et al.* Detection of immunogenic cell death and its relevance for cancer therapy. *Cell Death Dis* **11**, 1013 (2020).
10. Park, J.H. *et al.* Statin prevents cancer development in chronic inflammation by blocking interleukin 33 expression. *Nat Commun* **15**, 4099 (2024).
11. Sefried, S., Haring, H.U., Weigert, C. & Eckstein, S.S. Suitability of hepatocyte cell lines HepG2, AML12 and THLE-2 for investigation of insulin signalling and hepatokine gene expression. *Open Biol* **8** (2018).
12. Memon, A., Pyao, Y., Jung, Y., Lee, J.I. & Lee, W.K. A Modified Protocol of Diethylnitrosamine Administration in Mice to Model Hepatocellular Carcinoma. *Int J Mol Sci* **21** (2020).
13. Jung, Y., Zhao, M. & Svensson, K.J. Isolation, culture, and functional analysis of hepatocytes from mice with fatty liver disease. *STAR Protoc* **1**, 100222 (2020).

14. Zhu, Q. & Kanneganti, T.D. Cutting Edge: Distinct Regulatory Mechanisms Control Proinflammatory Cytokines IL-18 and IL-1 β . *J Immunol* **198**, 4210-4215 (2017).
15. Byun, J.S., Suh, Y.G., Yi, H.S., Lee, Y.S. & Jeong, W.I. Activation of toll-like receptor 3 attenuates alcoholic liver injury by stimulating Kupffer cells and stellate cells to produce interleukin-10 in mice. *J Hepatol* **58**, 342-349 (2013).
16. Udden, S.N. *et al.* NLRP12 suppresses hepatocellular carcinoma via downregulation of cJun N-terminal kinase activation in the hepatocyte. *Elife* **8** (2019).
17. Kurma, K. *et al.* DEN-Induced Rat Model Reproduces Key Features of Human Hepatocellular Carcinoma. *Cancers (Basel)* **13** (2021).
18. Lotze, M.T. & Tracey, K.J. High-mobility group box 1 protein (HMGB1): nuclear weapon in the immune arsenal. *Nat Rev Immunol* **5**, 331-342 (2005).
19. Chen, R., Kang, R. & Tang, D. The mechanism of HMGB1 secretion and release. *Exp Mol Med* **54**, 91-102 (2022).
20. Pittala, S., Krelin, Y. & Shoshan-Barmatz, V. Targeting Liver Cancer and Associated Pathologies in Mice with a Mitochondrial VDAC1-Based Peptide. *Neoplasia* **20**, 594-609 (2018).
21. Cheng, C. *et al.* Gehua Jiecheng Decoction Inhibits Diethylnitrosamine-Induced Hepatocellular Carcinoma in Mice by Improving Tumor Immunosuppression Microenvironment. *Front Pharmacol* **11**, 809 (2020).
22. Golino, J.L. *et al.* Anti-Cancer Activity of Verteporfin in Cholangiocarcinoma. *Cancers (Basel)* **15** (2023).
23. Xu, L. *et al.* The IL-33-ST2-MyD88 axis promotes regulatory T cell proliferation in the murine liver. *Eur J Immunol* **48**, 1302-1307 (2018).

Response to Reviewers' Comments:

We thank the reviewers for their insightful comments, which have improved our manuscript. Our replies to the comments are provided in blue below. Manuscript text and figures have also been revised (highlighted in blue) according to the reviewers' comments.

REVIEWER COMMENTS

Reviewer #1:

The authors adequately addressed the comments raised by the reviewers, and the contents have significantly improved.

We thank the reviewer for the positive comment.

Reviewer #2:

Post review, it is evident that Huang et al., have extensively addressed the points made by each of the reviewers. Overall, this study improves our understanding of HBV-associated HCC and identifies a ST2-Treg – IL-33 axis which exacerbates disease. The results suggest that inhibition of IL-33 signalling could provide therapeutic benefit in HBV-associated HCC.

Responses to comments:

Clarity and context:

The modifications to the manuscript have improved the contextualisation of these results within the existing literature and with reference to human disease. Furthermore, these changes have clarified the main points made by this study and now the conclusions drawn are more comprehensively supported by the data presented.

Limitations of the model used:

The addition of a section to focus on limitations of this study has improved the discussion and transparency of the work presented in this manuscript.

Addition of experiments:

The human scRNA-seq dataset showing an increase in ST2+ Treg in HBV-associated chronic hepatitis strengthens the data presented in the mouse model as both suggest the importance of this specific Treg population.

The additional quantification of liver IL-1 β , IL-4, and IL-16 has improved the rationale for focusing on IL-33 as it is clear now that it is altered in this model with respect to controls. The T cell culture using liver Treg instead of splenic has improved the relevance of the results in figure 2. This is in agreement with existing literature suggesting that tissue Tregs will exhibit greater expression of the functional markers such as ST2 in comparison with their splenic counterparts.

The additional experimental evidence provided clarifies the points made and solidifies the rationale of this study.

Flow cytometry:

Following the response to reviewer 2, point 11 (with reference to flow cytometry). The authors have satisfactorily addressed the concerns.

It would be important to note for future experiments that although Percoll has been used to isolate cells and it is understood that dead cells will be removed with this processing, it is still always better to add a viability stain. This is because after Percoll washing is required prior to staining and small numbers of cells can die during the normal staining process. It was also stated in the methods that a cell-activator cocktail is used for restimulation of the cells. With restimulation there will also always be more cell death.

Statistical Analysis:

The addition of further details concerning the statistical tests used, exact group sizes and number of independent experiments has clarified the robust nature of the study and improved transparency for the reader.

We thank the reviewer for the positive comments.

Reviewer #3:

In the amended version of Huang et al manuscript's, they report on the impact of HBV on chemical-induced liver carcinogenesis (DEN), through secretion of HMGB1. This DAMP led to the activation of IL-33, supporting the stimulation of ST2+ Treg cells and subsequently, the expression of TGF- and IL-10, which enhanced liver cancer development. Huang et al also focused on the protective effect of Pitavastatin (targeting the IL-33/Treg axis), which reduced the sensitivity to liver carcinogenesis in HBV-expressing DEN-treated mice and in patients. Translational aspect was also investigated by the quantification of ST2+ Tregs (scRNA seq data) in liver and IL-33 in serum samples from HBV-infected patients. Despite the appreciated considerable efforts made by Huang et al, to answer the several questions raised by the 3 reviewers and thus improve this amended version, some points remained unresolved.

We thank the reviewer for their constructive comments.

1) These authors claimed that HBV genome, transferred by a transfection method using hydrodynamic injection (HDI), in male mice at 1 month of age, remained persistent in the model of liver carcinogenesis (2-months of DEN injection). This viral persistence, previously illustrated by the stable level of HBsAg expression in untreated mice, is now also illustrated by HBc staining and quantification of HBV viral load (at 4 months post-transfection) in DEN-treated mice. However, these data have always been insufficient to validate the role of HBV viral persistence in the process of carcinogenesis and immune recruitment.

Indeed:

- The level of HBsAg was not illustrated in the blood of DEN-treated mice.

As requested, we confirmed HBsAg detection in the blood of HBV+DEN and HBV+PBS-treated WT mice at 8 months post-infection (Figure R1). We have added this data to our revised manuscript (Supplementary Fig. 1d).

Figure R1. Persistence of HBV protein expression after pAAV-HBV hydrodynamic injection and liver cancer induction by DEN treatment. Serum HBsAg levels in HBV+DEN- (n = 4) and HBV+PBS-treated (n = 7) WT mice at 8 months after hydrodynamic injection of pAAV-HBV plasmid. Each dot represents a mouse. Graph shows mean + SD, two-sided unpaired *t*-test.

- The amount of HBV DNA, quantified in the serum samples of DEN-treated mice, was reported as a relative value (vs. positive control of the kit used) and not as viral load. Quantification in IU/ml gives a more accurate assessment of the level of viral expression (as illustrated for patients in Supplementary Table 1).

We appreciate this valuable point. HBV DNA levels in mouse serum were quantified using the HBV TaqMan PCR Kit (NORGEN BIOTEK, Canada, Catalog no. TM29250), which is specifically designed for the detection of HBV DNA via real-time PCR using TaqMan technology. Due to the absence of a quantified positive control for HBV DNA, we are only able to determine the relative HBV concentration in comparison to the positive control provided in the kit. We have clarified this point in our revised manuscript (Materials and Methods section, page 20, first paragraph).

- Similarly, the number of HBc positive cells in the liver could be quantified for comparison to hepatic PCNA, IL-33 and HMGB1 staining.

As requested, we quantified Hepatitis B core antigen (HBcAg) positive cells in the liver of the mice (Figure R2). Considering that HBcAg is completely absent in Sham+DEN group, comparison with PCNA, IL-33, and HMGB1 staining was not possible. We have added this data to our revised manuscript (Supplementary Fig. 1g).

Figure R2. Hepatitis B core antigen (HBcAg) expression in the liver after pAAV-HBV hydrodynamic injection. Quantification of HBcAg⁺ cells per high power field (HPF) in WT mice at 4 months after the hydrodynamic injection of pAAV-HBV plasmid. HBcAg⁺ cells were counted in 5 to 10 randomly selected HPFs per liver. n = 4 mice in HBV+DEN group, n = 3 mice in Sham+DEN group, n = 3 mice in HBV+PBS group, and n = 3 mice in Sham+PBS group. Each dot represents one HPF. Graph shows mean + SD, one-way ANOVA with Tukey's multiple comparison test.

- Finally, both additional virological data in this amended manuscript were performed only at 4-months post-transfection (without liver tumor). The same analyses might be completed at 8 months.

We appreciate this point and have provided data supporting the persistence of HBsAg at 8 months post-infection (Figure R1 above).

2) As previously suggested, transfection using hydrodynamic injection maintains the plasmid genome mainly in episomal form into the nucleus of liver cells. Huang et al showed that 20 to 80% of hepatocytes were positive for PCNA and/or IL_33 and/or HMGB1 staining at 4-months post-HDI. In this context, episomal HBV DNA could probably be rapidly eliminated from hepatocytes and, considering the absence of HBV spreading, the link between HBV and immune regulation is uncertain throughout the carcinogenesis process.

We thank the reviewer for this insightful comment. As shown above and in Supplementary Figure 1, HBV DNA and gene expression persist in carcinogen-treated mice up to 8 months after pAAV-HBV hydrodynamic injection. We certainly agree that HBV level decreases over time due to the lack of infectious virions in this model. However, we propose that the persistence of the HBV genome during carcinogen treatment and early liver cancer development is essential for cancer promotion by inducing the DAMP/IL-33 axis in the liver. We have clarified this point and the limitation of our experimental model in the revised manuscript (Discussion section, page 13, second paragraph).

3) Park et al reported an increase of HMGB1 staining in the liver of HBV+DEN treated mice. This data was reinforced in this amended manuscript by the quantification of HMGB1 in liver and serum samples of treated mice. However, the serum quantification of this DAMP did not seem to be significantly different in presence or not of HBV in DEN-treated mice.

We appreciate this point. In our model, the liver is the source of HGMB1 induction in HBV+DEN-treated mice. We have clearly demonstrated the significantly increased levels of HGMB1 in HBV+DEN- compared with Sham+DEN-treated liver using immunostaining and measurement in liver lysates. HMGB1 levels in the serum of HBV+DEN-treated mice show a higher trend compared with Sham+DEN-treated mice. Nonetheless, HMGB1 serum levels may be impacted by local binding and consumption in the liver, liver size, and other factors.

4) Finally, could the contribution of HBV to the “modulating the immune response to environmental carcinogens” be related only to the early stages of HCC?

We completely agree with the reviewer. In our studies, we have focused on the mechanism by which HBV contributes to early HCC development. We have discovered that HBV modulates immune response (i.e., inflammation) upon exposure to carcinogens, which leads to tumor promotion in the liver. We have revised our manuscript to clarify this point (Discussion section, page 11, first paragraph).

5) Despite the better description of the model used by these authors, “HBV infection” is still mentioned in the amended version of this manuscript (such as line 101, 102, 113, p6....) and should be replaced by HBV expression or HBV infection-like, e.g.

We thank the reviewer for raising this point. We have revised the manuscript to avoid stating “HBV infection”.